# Reinforcement Learning in the Presence of Epistemic Ambivalence

## Abstract

The complexity of online decision-making under uncertainty stems from the requirement of finding a balance between exploiting known strategies and exploring new possibilities. Naturally, the uncertainty type plays a crucial role in developing decision-making strategies that manage complexity effectively. In this paper, we focus on a specific form of uncertainty known as *epistemic ambivalence* (EA), which emerges from conflicting pieces of evidence or contradictory experiences. It creates a delicate interplay between uncertainty and confidence, distinguishing it from epistemic uncertainty that typically diminishes with new information. Indeed, ambivalence can persist even after additional knowledge is acquired. To address this phenomenon, we propose a novel framework, called the *epistemically ambivalent Markov decision process* (EA-MDP), aiming to understand and control EA in decision-making processes. This framework incorporates the concept of a quantum state from the quantum mechanics formalism, and its core is to assess the probability and reward of every possible outcome. We calculate the reward function using quantum measurement techniques and prove the existence of an optimal policy and an optimal value function in the EA-MDP framework. We also propose the EA-epsilon-greedy Q-learning algorithm. To evaluate the impact of EA on decision-making and the expedience of our framework, we study two distinct experimental setups, namely the two-state problem and the lattice problem. Our results show that using our methods, the agent converges to the optimal policy in the presence of EA.

## 1 Introduction

Reinforcement Learning (RL) models an agent facing the exploration-exploitation dilemma: While being uncertain about some specific factors that determine the actions' outcomes, the agent selects between an action that, according to her current belief, maximizes the reward and a seemingly suboptimal one, which, however, can result in gaining some information that increases future rewards. The type of uncertainty plays a crucial role in solving the exploration-exploitation dilemma. Thus, the core challenge is to efficiently understand the environmental uncertainties. So far, the literature mainly emphasizes two primary forms of uncertainty: *Aleatoric uncertainty* and *epistemic uncertainty* Liu et al. (2024); Lockwood & Si (2022); Kahn et al. (2017); Lütjens et al. (2019), described below.

- Aleatoric uncertainty arises from the inherent randomness or stochastic nature of the environment in which the agent operates, such as randomness in reward generation processes, or state transitions Zennaro & Jøsang (2020); Clements et al. (2019); as such, it can neither be reduced nor predicted before starting an experiment.

- Epistemic uncertainty comes from the agent's limited knowledge like incomplete understanding of the environment or an inadequate representation of some of the characteristics of the problem. This uncertainty includes various aspects of the RL problem, including uncertainty regarding the true model of the environment, insufficient knowledge about the outcomes of specific actions in certain states, and ambiguity concerning the optimal policy due to incomplete exploration of the state

space Liu et al. (2024). The agent can reduce epistemic uncertainty if relevant information becomes available.

Effective decision-making necessitates that the agent adeptly manages both aleatoric and epistemic uncertainties, as this enhances performance and leads to more informed decisions in RL.

Besides aleatoric and epistemic uncertainties, other obstacles might hinder optimal decision-making. One is *ambivalence*, which can inhibit the decision-making progress. Ambivalence differs from uncertainty in that it persists even after the information becomes available. Consequently, traditional approaches to modeling, quantifying, and addressing uncertainty may be inadequate or suboptimal in effectively managing ambivalence as a distinct challenge Lam & Sherman (2020). In this work, we consider a novel form of uncertainty, called Epistemic Ambivalence (EA). EA occurs when multiple interpretations, explanations, or courses of action are possible, particularly in cases where the available data does not clearly support one viewpoint over another. In epistemology, which studies the nature and boundaries of knowledge, EA highlights the inherent limitations and complexities involved in understanding and reasoning about the world Amaya (2021); Williamson (2021); Lam (2013). While traditional uncertainty represents imperfect knowledge, EA represents a cognitive state in which conflicting pieces of evidence coexist, resulting in an uncertain decision-making process. Individuals may experience uncertainty or skepticism when supplied with contrasting or ambiguous information. For example, in investment decisions, a person may face conflicting information about a specific stock, which leads to EA. Even after performing extra research and gathering additional data, the person may remain uncertain about whether to buy or sell the stock because of the ongoing presence of conflicting evidence. Such examples illustrate how, when faced with contradictory data from credible sources, an agent may struggle to make an optimal decision, reflecting the nature of EA. The persistence of this uncertainty challenges traditional decision-making intuitions, highlighting the complexity of navigating conflicting evidence. Thus, a deeper understanding of EA can foster more flexible and adaptive decision-making strategies in complex and uncertain environments.

In quantum mechanics, quantum particles can exist in multiple states at the same time because of the principle of superposition, with their state remaining indeterminate until observed Nielsen & Chuang (2010). This principle makes an interesting connection to decision-making scenarios, where agents face uncertainty caused by conflicting information from diverse sources.

In this paper, we exploit the analogy between quantum mechanical states and EA to develop a theoretical framework for decision-making under EA and a policy. Roughly stated, we map each piece of information to a quantum basis state and construct an EA quantum state that incorporates all of the information. This mapping, accompanied by using Dirac notation and mathematical formulations derived from quantum mechanics, assists the decision-makers in quantifying EA, i.e., calculating the probability of each option, hence enabling informed and effective decision-making. We propose the EA-MDP framework to understand and control EA. We show that the optimal policy exists within this framework. Additionally, we introduce the EA-epsilon-greedy Q-learning algorithm to evaluate the impact of EA on decision-making. Two experiments are conducted to evaluate the performance and effectiveness of our approach.

The structure of the paper is as follows. In Section 2, we provide an overview of the relevant literature on uncertainty in RL, its application in quantum mechanics, and vice versa. Section 3 provides a brief introduction to the principles of quantum mechanics and superposition. Section 4 outlines the problem formulation and implementation of EA by using quantum mechanics. In Section 5, we present the theoretical results. Section 6 details the experimental evaluation using two distinct problems, where the agent's reward is calculated based on EA uncertainty, followed by the training process of the agent. Finally, Section 7 provides concluding remarks.

## 2 Related Work

Uncertainty exists in various elements of RL, including observations, actions, transition dynamics, and rewards Kakade & Langford (2002); Puterman (2014); Boutilier et al. (2000); Shapiro et al. (2021); Bertsekas & Tsitsiklis (1996); Wang et al. (2016). Wang et al. (2020) developed a robust reinforcement learning framework that allows agents to learn in environments with noise, where they can only observe perturbed

rewards. Zhang et al. (2020) studied the vulnerability of Deep Reinforcement Learning (DRL) agents to adversarial attacks on state observations. The authors discovered that a robust policy significantly enhances DRL performance in various environments, even in the absence of an adversary. Engel et al. (2005) used Gaussian processes to model uncertainty in rewards, enhancing the agent's ability to learn from sparse data. Kumar et al. (2020) introduced conservative Q-learning, which addresses uncertainty in rewards by penalizing the overestimation of Q-values in offline learning.

Quantum mechanics can enhance RL by using quantum phenomena such as superposition, entanglement, and quantum parallelism. These features allow quantum algorithms to process and explore multiple states or actions simultaneously. It has the potential to accelerate the learning process Biamonte et al. (2017); Dunjko et al. (2016); Dunjko & Briegel (2017); Saggio et al. (2021). An early paper by Dong et al. (2008) suggested that quantum mechanics can enhance learning processes. Neukart et al. (2018) showed that the optimization of strategies and the efficiency of learning are improved by the implementation of quantum-enhanced reinforcement learning in finite-episode games. Dalla Pozza et al. (2022) extended the classical concept of RL to the quantum domain in the lattice problem and investigated its behavior in a noisy environment. Dong et al. (2010) proposed a Quantum-inspired Reinforcement Learning (QiRL) algorithm for navigation control of autonomous mobile robots.

The use of quantum mechanics in decision-making has rapidly grown in recent years. The epistemic perspective of quantum states has been deeply investigated and has shown how to interpret quantum states as states of knowledge Fraser et al. (2023); Pusey et al. (2012); Caves et al. (2002); Healey (2017); Spekkens (2007); Leifer (2014). Busemeyer & Bruza (2012) developed quantum cognitive models that explain order effects, ambiguity, and sure-thing principle violations. Bruza et al. (2015) showed the superiority of quantum models over classical approaches in cases where incompatible events require a sequential evaluation. In these scenarios, quantum probability effectively captures human judgments under uncertainty and conflict. Ashtiani & Azgomi (2015) provided a comprehensive review of how quantum theory has been used to model decision-making and cognitive processes, especially in situations where classical probability fails. Helland et al. (2021) identified an epistemic connection between statistical inference and quantum theory. Helland (2023) used the Hilbert-space formalism for quantum decision theory and justified the Born rule via theoretical accessibility. Pothos & Busemeyer (2009) used quantum mechanics to explain sure-thing violations in gambling and the Prisoner's Dilemma. Yukalov & Sornette (2011) used the entanglement in quantum decision theory. Later, he unified quantum measurements and decision processes into one framework Yukalov & Sornette (2016). In a similar approach, ambivalence (such as emotional ambivalence, where opposing feelings like joy and sadness coexist) can be modeled as a superposition of quantum states Kyriazos & Poga (2025); Zoppolat et al. (2022). Favre et al. (2016) applied the quantum decision theory to an empirical dataset of binary lotteries to choose between a certain and a risky option. Widdows et al. (2023) used quantum circuits as initial quantum implementations of quantum cognition in a quantum hardware.

EA has not been previously investigated in the literature related to artificial intelligence. Therefore, in this paper, we introduce EA in this field, specifically in RL and quantum RL. The algorithm proposed in this paper uses quantum-inspired principles within a fully classical framework to model EA and support decision-making. By simulating quantum-like behavior, it avoids quantum hardware constraints such as the no-cloning theorem, decoherence, and noise in quantum computers.

## 3    Basics of Quantum Mechanics

In quantum mechanics, the *Dirac* notation, also called *bra-ket* notation, describes vectors and operators in Hilbert space Nielsen & Chuang (2010). Hilbert space represents a mathematical framework that generalizes the notation of Euclidean vector space to a finite or infinite dimensional space. The dimension of the Hilbert space depends on the specific system being studied. The state of a quantum system is represented by a vector state in a Hilbert space, denoted by *ket* $|\psi\rangle \in \mathcal{H}$. The vector state contains all the information about the system that is known.
The quantum system can exist in multiple states simultaneously. This phenomenon, referred to as the *superposition*, is a fundamental principle in quantum mechanics. Mathematically, it is expressed as a linear

combination of basis states $\{|j\rangle\}$,

$$|\psi\rangle = \sum_j c_j |j\rangle, \tag{1}$$

where the coefficients $c_j \in \mathbb{C}$ are complex probability amplitudes and $\sum_j |c_j|^2 = 1$. $|c_j|^2$ represents the probability of the quantum state $|\psi\rangle$ being in the basis state $|j\rangle$.

In quantum theory, complex probability amplitudes encode both magnitude and phase information, enabling the description of phenomena like superposition and interference. The squared magnitude of a probability amplitude gives the probability of an outcome, while the phase determines how different quantum states interfere, leading to effects such as constructive or destructive interference Ballentine (1970). Thus, quantum probability is fundamentally distinct from classical probability because it emerges from the intrinsic uncertainty of quantum states. *bra* $\langle\psi| = |\psi\rangle^\dagger$ shows the conjugate transpose of ket $|\psi\rangle$, with $\dagger$ being the conjugate transpose operation. The inner and outer products are well-defined in the Hilbert space:

- The **inner product** between two quantum states $|\psi\rangle$ and $|\phi\rangle$ is denoted as $\langle\phi|\psi\rangle$ and is a complex number. It quantifies the degree of overlap or projection between two quantum states. When the quantum state of the system is $|\psi\rangle$, the probability of measuring the system in the quantum state $|\phi\rangle$ is given by the magnitude squared of the inner product, $|\langle\phi|\psi\rangle|^2$.

- The **outer product** of two states, $|\psi\rangle$ and $|\phi\rangle$ is denoted by $|\phi\rangle\langle\psi|$. This operator maps $|\psi\rangle$ to $|\phi\rangle$ and can be used in the construction of projectors and other operators.

In quantum mechanics, operators $\boldsymbol{A} : \mathcal{H} \to \mathcal{H}$ represent physical observations and quantum state transformations. Hermitian operators satisfy $\boldsymbol{O}^\dagger = \boldsymbol{O}$ and have real eigenvalues. Unitary operators satisfy $\boldsymbol{U}^\dagger \boldsymbol{U} = \boldsymbol{U}\boldsymbol{U}^\dagger = \boldsymbol{I}$, where $\boldsymbol{I}$ is an identity operator. In this paper, bold capital letters are used to represent the operators.

To extract information from a quantum state, it is necessary to measure it using an observable quantity. During the measurement process, the quantum state collapses into one of the eigenstates of the measurement operator (i.e., observable quantity). The result obtained after measuring the quantum system is called the outcome. It is essential to perform the measurements several times since the outcome is probabilistic and it is not possible to accurately anticipate the exact outcome of one individual measurement. Nevertheless, one can use the expectation value to estimate the statistical distribution of the measurement results, as given by the Born rule Griffiths & Schroeter (2018). The expectation value of an observable $\boldsymbol{O}$ for the quantum state $|\psi\rangle$ is represented as

$$\langle\boldsymbol{O}\rangle_\psi = \langle\psi|\boldsymbol{O}|\psi\rangle. \tag{2}$$

This formula represents the statistical average of the measurement results related to the observable $\boldsymbol{O}$ while the system is in the state $|\psi\rangle$.

## 4  Problem Formulation

A Markov Decision Process (MDP) is a tuple $\mathcal{M} = \langle \mathcal{S}, \mathcal{A}, R, p, \gamma \rangle$, where $\mathcal{S}$ is the set of states, $\mathcal{A}$ is the set of actions, and $R$ is the reward function. In addition, $p$ represents the state transition distribution and $\gamma$ is a discount factor discounting the long-term rewards. To combine MDP and EA, we begin by defining the state space $\tilde{\mathcal{S}}$ and the action space $\tilde{\mathcal{A}}$ in a finite MDP in the presence of EA, and we refer to the resulting framework as EA-MDP. In an EA-MDP, the state space of an agent that interacts with the environment is defined as $\mathcal{H}_{\tilde{\mathcal{S}}} = \mathcal{H}_{\mathcal{S}} \otimes \mathcal{H}_{\text{EA}}$, where $\mathcal{H}_{\mathcal{S}}$ is the Hilbert space without any EA, and $\mathcal{H}_{\text{EA}}$ corresponds to the Hilbert space with EA, with dimensions $|\mathcal{H}_{\mathcal{S}}| = n$ and $|\mathcal{H}_{\text{EA}}| = m$. The symbol $\otimes$ represents the tensor product between two quantum states. The tensor product of quantum states is equivalent to their Kronecker product. We denote a quantum state by $|\tilde{s}(t)\rangle \in \mathcal{H}_{\tilde{\mathcal{S}}}$. At each time step $t$, the agent receives a representation of the quantum state of the environment $|\tilde{S}(t)\rangle \in \mathcal{H}_{\tilde{\mathcal{S}}}$, which we denote by

$$|\tilde{s}(t)\rangle = |s_t\rangle \otimes |\psi_{s_t}(t)\rangle, \tag{3}$$

where $|s_t\rangle$ is the underlying state of the system at time $t$ and $|\psi_{s_t}(t)\rangle$ is the *epistemic ambivalent quantum state* (EA quantum state) corresponding to the state $s_t$. The states $|s_t\rangle$ and $|\psi_{s_t}(t)\rangle$ are elements of the Hilbert spaces $\mathcal{H}_{\mathcal{S}}$ and $\mathcal{H}_{\text{EA}}$, respectively. The state $|\psi_s(t)\rangle$ can be defined in various ways based on the model of the environment. For example, it can be defined as a superposition of $m$ different EA bases, as follows.

$$|\psi_s(t)\rangle = \sum_{j=0}^{m-1} c_{s,j}(t)|j\rangle, \tag{4}$$

where $|c_{s,j}(t)|^2$ is the probability of the EA quantum state $|\psi_s(t)\rangle$ being at EA basis state $|j\rangle$ at time $t$.

In Equation (3), we consider two subsystems for a given quantum system. One for the underlying states and another one for the EA quantum states. In general, the underlying state can serve as a classical description of certain aspects of the environment, such as restricting the agent to occupy only one classical state at each time step. The representation given in Equation (3) is crucial for integrating classical and quantum aspects within the framework of EA-MDPs.

We enhance the notation of each quantum state by considering its underlying state. Therefore, we use $\tilde{\boldsymbol{s}}(s_t, t) \equiv |\tilde{s}(t)\rangle$ to denote the quantum state that contains the state $s_t$ with corresponding EA quantum state. In the current formulation, with a slight abuse of notation, we assume $\tilde{\boldsymbol{s}}(\cdot)$ acts as a function that takes a state $s_t \in \mathcal{S}$ and time $t$ as the input and returns the quantum state corresponding to state $s_t$ at time $t$.

**Assumption 1.** *In our problem formulation, we assume that each quantum state contains only a time-independent EA quantum state. Note that the quantum state itself remains time-dependent that reflects the time-dependent transitions of underlying states of the environment. In other words, $\tilde{\boldsymbol{s}}(\cdot)$ is a stationary function with respect to the input $t$.*

Based on Assumption 1, for all times $t$, the EA quantum state $|\psi_s(t)\rangle$ corresponding to a given state $s$ does not vary over time, which in turn implies that $c_{s,j}(t) = c_{s,j}$. Thus, for a given $s \in \mathcal{S}$, we have $|\psi_s(t)\rangle = |\psi_s(t')\rangle, \forall t, t'$. Therefore, in the following, we drop the index $t$ in the notation of EA quantum state and use $|\psi_s\rangle$ instead. Similarly, we drop the corresponding time index in the notation of quantum states, i.e., $\tilde{\boldsymbol{s}}(s) \equiv |\tilde{s}\rangle$. An interesting research direction is to consider the dynamic case where EA quantum states evolve themselves, i.e., the operator $|\psi_s(t)\rangle$ being dependent on time $t$. For now, we continue with the simpler version of this problem, where EA quantum states remain fixed for each state $s$ of the environment throughout the play.

We formally define EA-MDP as a tuple $\tilde{\mathcal{M}} = \langle \tilde{\mathcal{S}}, \tilde{\mathcal{A}}, r, p, \gamma \rangle$, where $\tilde{\mathcal{A}}$ denotes the set of actions and $r$ represents the reward function for quantum states. We use $\tilde{\mathcal{A}}(\tilde{\boldsymbol{s}}(s))$ to represent the available actions for a given quantum state $\tilde{\boldsymbol{s}}(s)$. However, to simplify the notation, we consider a general set of actions $\tilde{\mathcal{A}}$ for all quantum states, i.e., $\tilde{A}(\tilde{\boldsymbol{s}}(S_t)) \equiv \tilde{A}_t$. At each time $t$, given $\tilde{\boldsymbol{s}}(S_t)$, the agent selects an action $\tilde{A}_t \in \tilde{\mathcal{A}}$. Then, the agent receives a numerical reward $r_{t+1} \subset \mathbb{R}$ based on the reward function $r$, and the environment's quantum state evolves into a new quantum state $\tilde{\boldsymbol{s}}(S_{t+1})$. The trajectory of the agent is represented by

$$\tilde{\boldsymbol{s}}(S_0), \tilde{A}_0, r_1, \tilde{\boldsymbol{s}}(S_1), \tilde{A}_1, r_2, \ldots. \tag{5}$$

It is necessary to comprehend how the environment can provide rewards when an agent takes action. The rewards can be paid after measuring the quantum state using a reward operator $\boldsymbol{R}$. Due to the probabilistic nature of a single measurement's outcome, it is essential to perform many measurements, calculate the expectation value of the reward, and use it as the reward function. In this paper, we make an additional assumption while calculating the expectation value of the reward.

**Assumption 2.** *For simplicity, we assume that the reward depends only on the next quantum state $\tilde{\boldsymbol{s}}(s')$, to which the environment transitions after taking action on the current quantum state $\tilde{\boldsymbol{s}}(s)$. This assumption in quantum mechanics implies that we should measure the reward operator exclusively in the next quantum state.*

In this paper, we employ a quantum measurement technique known as projective measurements Nielsen & Chuang (2010). The outcome reward function, denoted as $\tilde{r} : \tilde{\Omega} \to \mathbb{R}$, assigns a reward for each measurement

outcome $|\tilde{\omega}\rangle \in \mathcal{H}_{\tilde{\mathcal{S}}}$, where we also use $|\tilde{\omega}\rangle$ to refer to the outcome within the text. Here, $\tilde{\Omega} = \{|\tilde{\omega}\rangle\}$ represents the finite set of complete and orthogonal outcomes. To determine the probability of measuring a particular outcome $|\tilde{\omega}\rangle$, we define a positive semi-definite operator, represented as $\boldsymbol{P}_{\tilde{\omega}} : \mathcal{H}_{\tilde{\mathcal{S}}} \to \mathcal{H}_{\tilde{\mathcal{S}}}$, where $\sum_{\tilde{\omega} \in \tilde{\Omega}} \boldsymbol{P}_{\tilde{\omega}} = \boldsymbol{I}$. The probability of measuring $|\tilde{\omega}\rangle$ given that the environment is in a quantum state $\tilde{\boldsymbol{s}}(s') \equiv |\tilde{s}'\rangle$ is

$$P_{\tilde{\omega}}\big(\tilde{\boldsymbol{s}}(s')\big) = \langle \tilde{s}' \,|\, \boldsymbol{P}_{\tilde{\omega}} \,|\, \tilde{s}'\rangle, \tag{6}$$

with $\sum_{\tilde{\omega}} P_{\tilde{\omega}}(\tilde{\boldsymbol{s}}(s')) = 1$. In the projective measurement $\boldsymbol{P}_{\tilde{\omega}} = |\tilde{\omega}\rangle \langle \tilde{\omega}|$, as a result, we have

$$P_{\tilde{\omega}}\big(\tilde{\boldsymbol{s}}(s')\big) = |\langle \tilde{\omega}|\tilde{s}'\rangle|^2. \tag{7}$$

**Definition 1.** *Given the quantum state $\tilde{\boldsymbol{s}}(s')$, the expectation value of the reward is given by*

$$r\big(\tilde{\boldsymbol{s}}(s')\big) \doteq \sum_{\tilde{\omega}} P_{\tilde{\omega}}\big(\tilde{\boldsymbol{s}}(s')\big)\tilde{r}(\tilde{\omega}). \tag{8}$$

The reward mechanism for EA-MDP can be comprehended through the joint probability

$$p\big(\tilde{\boldsymbol{s}}(s'), \tilde{r}(\tilde{\omega})|\tilde{\boldsymbol{s}}(s), \tilde{a}\big) = \Pr\Big\{\tilde{S}_t = \tilde{\boldsymbol{s}}(s'), R_t = \tilde{r}(\tilde{\omega})|\tilde{S}_{t-1} = \tilde{\boldsymbol{s}}(s), \tilde{A}_{t-1} = \tilde{a}\Big\}, \tag{9}$$

for all $s, s' \in \mathcal{S}$, $\tilde{\omega} \in \tilde{\Omega}$, and $\tilde{a} \in \tilde{\mathcal{A}}$. The function $p$ represents the probability of the reward $\tilde{r}(\tilde{\omega})$ received when the action $\tilde{a}$ is taken from the quantum state $\tilde{\boldsymbol{s}}(s)$ to the quantum state $\tilde{\boldsymbol{s}}(s')$. The function $p : \mathcal{S} \times \tilde{\Omega} \times \mathcal{S} \times \tilde{\mathcal{A}} \to [0, 1]$ satisfies

$$\sum_{s' \in \mathcal{S}} \sum_{\tilde{\omega} \in \tilde{\Omega}} p\big(\tilde{\boldsymbol{s}}(s'), \tilde{r}(\tilde{\omega})|\tilde{\boldsymbol{s}}(s), \tilde{a}\big) = 1, \forall s \in \mathcal{S}, \tilde{a} \in \tilde{\mathcal{A}}. \tag{10}$$

The probabilities provided by $p$ define the dynamics entirely of the environment in an EA-MDP. In MDP, the expected reward can be calculated as

$$r(s', a, s) = \mathbb{E}\Big\{R_t|S_{t-1} = s, A_{t-1} = a, S_t = s\Big\} = \sum_{r \in R} r\frac{p(s', r|s, a)}{p(s'|s, a)}. \tag{11}$$

By comparing (8) and (11) and determining the similarity between the probabilities of EA-MDP and MDP, we obtain

$$P_{\tilde{\omega}}\big(\tilde{\boldsymbol{s}}(s')\big) = \frac{p\big(\tilde{\boldsymbol{s}}(s'), \tilde{r}(\tilde{\omega})|\tilde{\boldsymbol{s}}(s), \tilde{a}\big)}{p\big(\tilde{\boldsymbol{s}}(s')|\tilde{\boldsymbol{s}}(s), \tilde{a}\big)}, \tag{12}$$

which also satisfies Equation (10). $p\big(\tilde{\boldsymbol{s}}(s')|\tilde{\boldsymbol{s}}(s), \tilde{a}\big)$ is the EA-MDP state transition distribution.

## 5 Theoretical Results

In EA-MDP, similar to traditional RL problems, the goal is to maximize the expected discounted return. The agent aims to find the optimal policy, which is a conditional distribution $\pi(\tilde{a}, \tilde{\boldsymbol{s}})$ that maximizes the value function. A stochastic policy is defined as a function $\pi : \tilde{\mathcal{S}} \to \Delta(\tilde{\mathcal{A}})$, where $\Delta(\tilde{\mathcal{A}})$ is the set of probability distributions over the action set $\tilde{\mathcal{A}}$. In an EA-MDP, we define the value function $V_{\pi}^{\tilde{\boldsymbol{s}}}(s)$ and the action-value function $Q_{\pi}^{\tilde{\boldsymbol{s}}}(s, \tilde{a})$ as

$$V_{\pi}^{\tilde{\boldsymbol{s}}}(s) \doteq \mathbb{E}_{\pi}\Big[G_t \mid \tilde{S}_t = \tilde{\boldsymbol{s}}(s)\Big], \tag{13a}$$

$$Q_{\pi}^{\tilde{\boldsymbol{s}}}(s, \tilde{a}) \doteq \mathbb{E}_{\pi}\Big[G_t \mid \tilde{S}_t = \tilde{\boldsymbol{s}}(s), \tilde{A}_t = \tilde{a}\Big], \tag{13b}$$

where $G_t = \sum_{k=0}^{\infty} \gamma^k r_{t+k+1} = r_{t+1} + \gamma G_{t+1}$ is the discounted cumulative reward.

**Theorem 1** (Bellman equations in EA-MDP). *In an EA-MDP, denoting as $\tilde{\mathcal{M}} = \langle \tilde{\mathcal{S}}, \tilde{\mathcal{A}}, r, p, \gamma \rangle$, with a fixed stochastic policy $\pi : \tilde{\mathcal{S}} \to \Delta(\tilde{\mathcal{A}})$ and a fixed $\tilde{\boldsymbol{s}}(\cdot)$, we have*

$$V_\pi^{\tilde{\boldsymbol{s}}}(s) = \sum_{\tilde{a} \in \tilde{\mathcal{A}}} \pi\big(\tilde{a}|\tilde{\boldsymbol{s}}(s)\big) \sum_{s' \in \mathcal{S}} p\big(\tilde{\boldsymbol{s}}(s')|\tilde{\boldsymbol{s}}(s), \tilde{a}\big) \Big[ r\big(\tilde{\boldsymbol{s}}(s')\big) + \gamma V_\pi^{\tilde{\boldsymbol{s}}}(s') \Big], \tag{14a}$$

$$Q_\pi^{\tilde{\boldsymbol{s}}}(s, \tilde{a}) = \sum_{s' \in \mathcal{S}} p\big(\tilde{\boldsymbol{s}}(s')|\tilde{\boldsymbol{s}}(s), \tilde{a}\big) \Big[ r\big(\tilde{\boldsymbol{s}}(s')\big) + \gamma \sum_{\tilde{a}' \in \tilde{\mathcal{A}}} \pi\big(\tilde{a}'|\tilde{\boldsymbol{s}}(s')\big) Q_\pi^{\tilde{\boldsymbol{s}}}(s', \tilde{a}') \Big]. \tag{14b}$$

*Proof.* See Section A.1 in the supplementary material. $\square$

**Theorem 2** (Bellman contraction for EA-MDP). *In an EA-MDP, denoted as $\tilde{\mathcal{M}} = \langle \tilde{\mathcal{S}}, \tilde{\mathcal{A}}, r, p, \gamma \rangle$, the Bellman operator $T$ for a value function is defined as*

$$\big(TV^{\tilde{\boldsymbol{s}}}\big)(s) = \max_{\tilde{a} \in \tilde{A}} \left[ \sum_{s' \in \mathcal{S}} p\big(\tilde{\boldsymbol{s}}(s')|\tilde{\boldsymbol{s}}(s), \tilde{a}\big) \left( r\big(\tilde{\boldsymbol{s}}(s')\big) + \gamma V^{\tilde{\boldsymbol{s}}}(s') \right) \right], \tag{15}$$

*and is a contraction mapping.*

*Proof.* See Section A.2 of the supplementary material. $\square$

The optimal policy $\pi^*$ in EA-MDP maximizes the value function for all states. Hence, the optimal value function $V_{\pi^*}^{\tilde{\boldsymbol{s}}}$ is the maximum value function when following the optimal policy $\pi^*$. Formally,

$$V_{\pi^*}^{\tilde{\boldsymbol{s}}}(s) \geq V_\pi^{\tilde{\boldsymbol{s}}}(s), \quad \forall s \in \mathcal{S} \text{ and } \forall \pi. \tag{16}$$

**Theorem 3** (Existence of an optimal value function and optimal policy in EA-MDP). *In an EA-MDP, denoted as $\tilde{\mathcal{M}} = \langle \tilde{\mathcal{S}}, \tilde{\mathcal{A}}, r, p, \gamma \rangle$, there exists an optimal value function $V_{\pi^*}^{\tilde{\boldsymbol{s}}}$ and at least one optimal policy $\pi^*$.*

*Proof.* See Section A.3 of the supplementary material. $\square$

Consider that instead of computing the exact expectation value of the reward, we estimate it by performing $n$ quantum measurements. In this case, the reward estimate is subject to statistical error. The resulting deviation between the true optimal value function and the learned value function is bounded by the following theorem.

**Theorem 4** (Suboptimality bound under finite-sample reward estimation). *Let $\hat{r}$ denote the empirical estimate of the expected reward obtained from $n$ independent quantum measurements, and let $\hat{\pi}^*$ be any policy that is optimal with respect to the estimated reward $\hat{r}$. Then, with probability at least $1 - \delta$, the following bound holds.*

$$\|V_{\hat{\pi}^*}^{\tilde{\boldsymbol{s}}} - V_{\pi^*}^{\tilde{\boldsymbol{s}}}\|_\infty \leq \frac{2\Delta}{1 - \gamma} \sqrt{\frac{\log(2/\delta)}{2n}}, \tag{17}$$

*where $\Delta$ denotes the difference between the maximum and minimum eigenvalues of the reward operator $\mathbf{R}$. This bound quantifies the suboptimality of the learned policy $\hat{\pi}^*$ with respect to an optimal policy $\pi^*$ under the true expectation value of the reward.*

*Proof.* See Section A.4 of the supplementary material. $\square$

**Remark 5** (Time-dependent EA quantum states). *As stated in Assumption 1, we assume in this work that EA quantum states are time-independent. This assumption is made for modeling simplicity and to keep the role of the EA layer in the reward structure as transparent as possible. Nevertheless, the framework can be extended to a time-dependent EA quantum state by allowing it to evolve under a unitary operator, $|\psi_s(t_2)\rangle = \boldsymbol{U}(t_2, t_1) |\psi_s(t_1)\rangle$. In addition, the current framework can be further extended to allow both the*

*measurement outcome $|\tilde{\omega}\rangle$ and the outcome reward function $\tilde{r}(\tilde{\omega})$ to be time-dependent. As long as the quantum state evolution is Markovian and the rewards remain bounded under a discount factor, the standard sup-norm contraction proof continues to hold. Therefore, our theoretical results extend naturally to the time-dependent EA setting.*

We would like to highlight that our model and solution allow the agent to tackle the EA uncertainty, which is encoded via probability amplitudes and the superposition of quantum states, by estimating the rewards using an extra expectation over the possible outcomes. Our model is a quantum-inspired model and differs from conventional MDPs with respect to the reward and value function definition as there is no outcome set in conventional MDPs. In EA-MDP, there is an extra expectation over outcomes which is the result of having probability amplitudes and an outcome set. This outcome set helps to assign rewards to the multiple configurations of conflicting evidence.

Due to space constraints, the rest of the theoretical results are moved to Sections C and D of the supplementary material. In Section C of the supplementary material, we calculate the reward operator. Additionally, in Section D of the supplementary material, we explain the method for computing the reward function when the environment provides the outcome reward solely based on the EA component of the quantum state.

## 6 Experimental Results and Analysis

To clarify more about the EA, we conduct two experiments using two toy models. In the initial experiment, we investigate a two-site toy model under the influence of EA. In the second experiment, we explore a multi-site system in which the agent must identify the optimal pathway within a square lattice to reach the terminal state in the presence of EA. In both experiments, we calculate the optimal value function and examine the quantum interference resulting from the complex probability amplitudes. These experiments will help us better understand how uncertainty and conflicting evidence impact decision-making in various scenarios. By analyzing the results, we can gain insights into how individuals navigate complex information to shape beliefs and make choices. For simplicity, in these experiments, we use a separated outcome in the EA basis, as fully explained in detail in Section D of the supplementary material. We consider the separable outcome of the form $|\tilde{\omega}\rangle = |s'\rangle \otimes |\omega^{(\mathrm{EA})}\rangle$ and the set of outcomes $\Omega^{(\mathrm{EA})} = \{|\omega^{(\mathrm{EA})}\rangle\}$, written in the EA basis. We assume that the outcome reward is only given for the EA components; in other words, $\tilde{r}(\tilde{\omega}) = \tilde{r}(\omega^{(\mathrm{EA})})$. In brief, our experiments in EA-MDP are defined as follows.

- A discrete set of underlying states
$$\mathcal{S} = \{s^1, s^2, \dots, s^n\},$$
where $s_t \in \mathcal{S}$ is the underlying state of the environment at time $t$. In this example, the underlying state represents the position of the agent in the lattice.

- A discrete set of EA bases
$$\{|0\rangle, |1\rangle, \dots, |m-1\rangle\}.$$
At each time step, the environment is at quantum state $\tilde{s}(S_t)$, which contains the underlying state and a linear combination of EA bases.

- A set of actions
$$\tilde{\mathcal{A}} = \{\tilde{a}^1, \tilde{a}^2, ..., \tilde{a}^k\},$$
where $\tilde{a}_t$ shows the action that the agent selects at time $t$.

- The set of transition functions $\mathcal{P} = \{p(\tilde{s}(s')|\tilde{s}(s), \tilde{a})\}$ that maps the quantum state $\tilde{s}(s)$ with the action $\tilde{a}$ to the next state $\tilde{s}(s')$ with a given transition probability. In the experimental result, the transition function is deterministic, meaning that $p(\tilde{s}(s')|\tilde{s}(s), \tilde{a}) = 0$ or $1$, $\forall s, \tilde{a}$.

- A reward function $r : \tilde{\mathcal{S}} \to \mathbb{R}$, which calculates the expectation value of the reward based on the next quantum state $\tilde{s}(s')$, the discrete set of EA outcomes
$$\Omega^{(\mathrm{EA})} = \{|\omega_0^{(\mathrm{EA})}\rangle, |\omega_1^{(\mathrm{EA})}\rangle, \dots, |\omega_{p-1}^{(\mathrm{EA})}\rangle\}, \tag{18}$$

and the separated outcome reward function $\tilde{r} : \Omega^{(\text{EA})} \to \mathbb{R}$. The reward is calculated with

$$r\big(\tilde{s}(s')\big) = \sum_{\omega^{(\text{EA})}} \tilde{r}(\omega^{(\text{EA})}) \left| \left\langle \omega^{(\text{EA})} \middle| \psi_{s'} \right\rangle \right|^2 \tag{19}$$

- A discount factor $0 \leq \gamma \leq 1$.

To improve readability, we collect probability amplitudes and outcome rewards in vectors and use vector notations $\boldsymbol{c}_i = (c_{i,0}, c_{i,2}, \ldots, c_{i,m-1})$ and $\tilde{\boldsymbol{r}} = (\tilde{r}(\omega_0^{(\text{EA})}), \tilde{r}(\omega_1^{(\text{EA})}), \ldots, \tilde{r}(\omega_{p-1}^{(\text{EA})}))$, respectively.

### 6.1 Model 1: Two-site System

To clarify the EA formulation of outcomes and reward, let us examine a system consisting of two sites. Concerning the action set, we assume that $\tilde{\mathcal{A}} = \{\rightleftarrows\}$ consists of only one action $\rightleftarrows$ which represents the process of moving from one site to another. Furthermore, let us suppose that we have a set of three orthogonal EA bases ($m = 3$) denoted as $\{|0\rangle, |1\rangle, |2\rangle\}$. The EA quantum state in the EA Hilbert space $\mathcal{H}_{\text{EA}}$ is defined as a linear combination of the basis states of EA, as represented by Equation (4). Consequently, for a given quantum state $\tilde{\boldsymbol{s}}(s)$, we have

$$\begin{aligned} \tilde{\boldsymbol{s}}(s_i) \equiv |\tilde{s}_i\rangle &= |s_i\rangle \otimes |\psi_{s_i}\rangle \\ &= |s_i\rangle \otimes \Big( c_{i,0} |0\rangle + c_{i,1} |1\rangle + c_{i,2} |2\rangle \Big). \end{aligned} \tag{20}$$

We consider the set of outcomes expressed in the EA as

$$\Omega^{(\text{EA})} = \left\{ \frac{|0\rangle + i|1\rangle}{\sqrt{2}}, \frac{|0\rangle - i|1\rangle}{\sqrt{2}}, |2\rangle \right\}. \tag{21}$$

The first and second outcomes consist of both $|0\rangle$ and $|1\rangle$, meaning they contribute to the reward for both $|0\rangle$ and $|1\rangle$ simultaneously, but with different probability amplitudes. Finally, as we have only two sites and the agent cannot stay in a fixed site for two consecutive rounds of play, the state transition function $p(\tilde{\boldsymbol{s}}(s_j)|\tilde{\boldsymbol{s}}(s_i), \tilde{a}_k)$ is a deterministic function.

With the setting mentioned above, let us calculate the value function for both the quantum states $\tilde{\boldsymbol{s}}(s_1)$ and $\tilde{\boldsymbol{s}}(s_2)$. By using Equation (14a) and assuming that the agent follows a stochastic policy $\pi(\cdot|\tilde{\boldsymbol{s}}(s))$, we observe that

$$V_\pi^{\tilde{\boldsymbol{s}}}(s_1) = \sum_{\tilde{a} \in \tilde{\mathcal{A}}} \pi(\tilde{a}|\tilde{\boldsymbol{s}}(s_1)) \Big[ r(\tilde{\boldsymbol{s}}(s_2)) + \gamma V_\pi^{\tilde{\boldsymbol{s}}}(s_2) \Big], \tag{22a}$$

$$V_\pi^{\tilde{\boldsymbol{s}}}(s_2) = \sum_{\tilde{a} \in \tilde{\mathcal{A}}} \pi(\tilde{a}|\tilde{\boldsymbol{s}}(s_2)) \Big[ r(\tilde{\boldsymbol{s}}(s_1)) + \gamma V_\pi^{\tilde{\boldsymbol{s}}}(s_1) \Big]. \tag{22b}$$

In this case, the jump action occurs between both sites over an infinite number of steps, and the policy is deterministic. Consequently, we can easily obtain the optimal value function as follows

$$V_{\pi^*}^{\tilde{\boldsymbol{s}}}(s_1) = \frac{r(\tilde{\boldsymbol{s}}(s_2)) + \gamma\left(r(\tilde{\boldsymbol{s}}(s_1))\right)}{1 - \gamma^2}, \tag{23a}$$

$$V_{\pi^*}^{\tilde{\boldsymbol{s}}}(s_2) = \frac{r(\tilde{\boldsymbol{s}}(s_1)) + \gamma\left(r(\tilde{\boldsymbol{s}}(s_2))\right)}{1 - \gamma^2}. \tag{23b}$$

The optimal value functions for constant probability amplitudes $\boldsymbol{c}_1$ and $\boldsymbol{c}_2$ are shown in Fig. 1. The relationship between the optimal value function and the discount factor $\gamma$ is illustrated in Fig. 1-(a), while the relationship with the outcome reward $\tilde{r}(\omega_2^{(\text{EA})})$ is shown in Fig. 1-(b). The results demonstrate that the computed values in Equation (23) are achieved in practice, confirming the existence of both an optimal value function and an optimal policy. When using complex probability amplitudes in Equation (20), quantum interference leads to constructive or destructive patterns, resulting in either an increase or decrease in the optimal value function. For a detailed discussion, we refer to Section F.1 of the supplementary material.

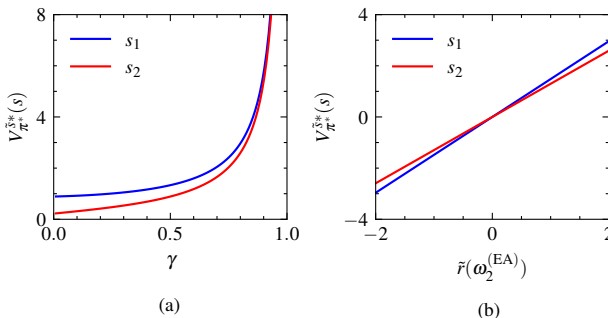

(a)

(b)

Figure 1: The optimal value function for the two-sites example in the presence of EA with parameters $\tilde{\boldsymbol{r}} = (-1, 1, \tilde{r}(\omega_2^{(\text{EA})}))$, $\boldsymbol{c}_1 = (\frac{2}{3}, \frac{2}{3}, \frac{1}{3})$, and $\boldsymbol{c}_2 = (\frac{2}{3}, \frac{1}{3}, \frac{2}{3})$. The set of outcomes is shown in Equation (21). (a) $\tilde{r}(\omega_2^{(\text{EA})}) = 2$ and different values of $\gamma$, (b) $\gamma = 0.8$ and different values of $\tilde{r}(\omega_2^{(\text{EA})})$.

## 6.2 Model 2: More Complex Example of Many-Site Systems

Consider a two-dimensional lattice of size $L_x \times L_y$ with some obstacles and one goal site (terminal state) inside it. An agent navigates through this lattice and attempts to collect the maximum reward in the presence of EA. Each site contains EA with different states, which are presented as EA quantum states. At each site, due to the presence of EA, the state has conflicting pieces of evidence. To clarify further, consider the $5 \times 5$ lattice shown in Fig. 2-(a). We consider an EA set of bases $\{|0\rangle, |1\rangle, |2\rangle, |3\rangle\}$. These bases appear with respective amplitudes at each site $i$ as $\mathbf{c}_i = (c_{i,0}, c_{i,1}, c_{i,2}, c_{i,3})$. Furthermore, all possible actions at each site of the lattice are represented in the respective site by vectors. Each action shows a possible move to the nearest neighbor site. In addition, there are two gray sites marked with $b$ that serve as obstacles, into which the agent is forbidden from entering. The colors in Fig. 2-(b) are EA bases and the combination of those bases (with different amplitudes) creates EA in each site. Time-independent EA quantum states can be constructed using Equation (4) at each site. As an example for this experiment, consider the EA

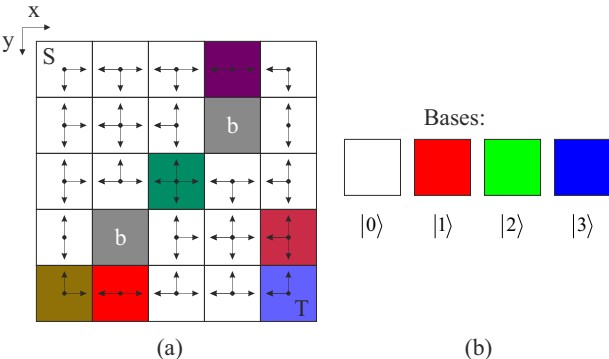

(a)

(b)

Figure 2: (a) A $5 \times 5$ lattice. Color combinations represent the sites with different EA quantum states. (b) The set of EA bases shown with different colors.

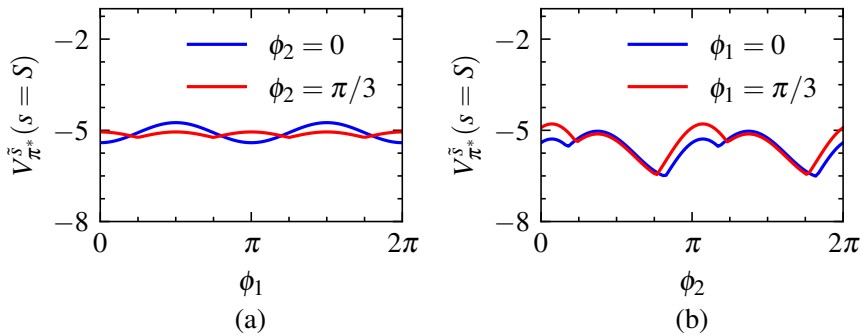

Figure 3: The optimal value function in the lattice, given an EA with outcome rewards $\tilde{\boldsymbol{r}} = (-1, -2, -3, 1)$, a discount factor $\gamma = 0.9$, probability amplitudes in Equation (24), and the set of outcomes in Equation (25). The effect of varying (a) $\phi_1$ and (b) $\phi_2$ on the optimal value function is shown.

quantum state for each site, with coordinates $(x, y)$, represented as

$$\left|\psi_{(4,1)}(t)\right\rangle = \frac{1}{\sqrt{2}}\left|1\right\rangle + \frac{1}{\sqrt{2}}\left|3\right\rangle, \tag{24a}$$

$$\left|\psi_{(3,3)}(t)\right\rangle = \frac{1}{\sqrt{2}}\left|2\right\rangle + \frac{i}{\sqrt{2}}\left|3\right\rangle, \tag{24b}$$

$$\left|\psi_{(5,4)}(t)\right\rangle = \frac{2}{5}\left|0\right\rangle + \frac{4}{5}\left|1\right\rangle + \frac{1}{5}\left|2\right\rangle + \frac{2}{5}\left|3\right\rangle, \tag{24c}$$

$$\left|\psi_{(1,5)}(t)\right\rangle = \frac{1}{\sqrt{2}}\left|1\right\rangle + \frac{i}{\sqrt{2}}\left|2\right\rangle, \tag{24d}$$

$$\left|\psi_{(5,5)}(t)\right\rangle = \frac{1}{\sqrt{5}}\left|0\right\rangle + \frac{1}{\sqrt{5}}\left|1\right\rangle + \frac{i}{\sqrt{5}}\left|2\right\rangle + \sqrt{\frac{2}{5}}\left|3\right\rangle, \tag{24e}$$

$$\left|\psi_{(others)}(t)\right\rangle = \left|0\right\rangle. \tag{24f}$$

We consider the outcome reward vector as $\tilde{\boldsymbol{r}} = (-1, -2, -3, \tilde{r}(\omega_3^{(\mathrm{EA})}))$, and a set of outcomes characterized by parameters $\phi_1$ and $\phi_2$, as follows

$$\Omega^{(\mathrm{EA})}(\phi_1, \phi_2) = \left\{ \frac{\cos\phi_1\left|0\right\rangle + i\sin\phi_1\left|1\right\rangle}{\sqrt{2}}, \frac{i\sin\phi_1\left|0\right\rangle + \cos\phi_1\left|1\right\rangle}{\sqrt{2}}, \right.$$
$$\left. \frac{\cos\phi_2\left|2\right\rangle + i\sin\phi_2\left|3\right\rangle}{\sqrt{2}}, \frac{i\sin\phi_2\left|2\right\rangle + \cos\phi_2\left|3\right\rangle}{\sqrt{2}} \right\}, \tag{25}$$

where $\phi_1$ and $\phi_2$ are adjustable parameters used to modify the outcomes.

In the lattice environment, the positive rewards encourage the agent to explore more and stay inside the lattice for a longer time. On the other hand, the negative rewards encourage the agent to leave the lattice as soon as possible. Our proposed algorithm, namely EA-epsilon-greedy Q-Learning (EA-QL), is outlined in Section B of the supplementary material (Algorithm 1). We apply EA-QL algorithm to determine the most beneficial path with the highest cumulative reward from the starting point to the terminal state. At the phase transition points, the optimal policy shifts to a new optimal policy, resulting in an adjustment in the trajectory in the lattice. In this scenario, the interference effect of the complex probability amplitudes disappears. Fig. 3 shows the effect of changing parameters $\phi_1$ and $\phi_2$ on the optimal value function. As the parameters $\phi_1$ and $\phi_2$ vary, the optimal value function changes due to interference effects. These effects potentially lead to changes in the optimal policy and the optimal path of the agent. When $\phi_1 = \phi_2 = 0$, the condition in Theorem 8, ($\left|\omega^{(\mathrm{EA})}\right\rangle = \left|j\right\rangle$), is fulfilled. Further experimental results are presented in Section F.2 of the supplementary material.

**Challenges and limitations:** Implementing our proposed EA-MDP framework in practice and applying it to real-world problems is a challenging task. The main challenge is to convert classical data into quantum data, particularly in the form of superposition. In recent years, the conversion of classical data into quantum data (quantum states) has gained significant attention Biamonte et al. (2017). Besides, a set of outcome representations is needed to enable such an implementation. This set of outcomes is essential for calculating the rewards, as it is used to determine the measurement and the reward. Selecting a suitable outcome set is crucial to ensure accurate reward calculations. In addition, in real-world experiments with non-separable outcome sets, the number and dimension of outcomes may increase significantly. If $n$ is the number of underlying states and $m$ is the number of EA bases, the memory needed to store the outcomes scales by a factor of $O((mn)^2)$. Similarly, the memory needed to store the separated quantum states scales by a factor of $O(mn)$. Furthermore, the computational time required to calculate the reward scales as $O((mn)^2)$. For large, non-separable outcome sets, such a fully dense representation becomes costly. However, several methods from quantum mechanics can keep the EA layer computationally feasible. We provide a brief overview of these techniques in Section E of the supplementary material.

**Reproducibility Statement:** We ensure the reproducibility of all the experiments presented in this paper. Detailed descriptions of the experiments, including key implementation steps, algorithmic procedures, and parameter settings, are provided in both the main text and the supplementary material. We utilized reference Sochorová & Jamriska (2021) to blend the colors correctly and applied them to the EA states in Fig. 2-(a). ChatGPT and QuillBot were used to polish the language of this paper.

## 7 Conclusion

In this paper, we address a specific type of uncertainty known as epistemic ambivalence, which arises from conflicting information or contradictory experiences. We introduced the EA-MDP framework, grounded in the principle of superposition from quantum mechanics, where a quantum particle can exist in multiple states simultaneously. The quantum state contains information about both the underlying state and the related EA, which is encoded with complex probability amplitudes. Using quantum measurement, we calculated the probability of given outcomes based on the quantum state. An interesting direction for future research is to extend the EA-MDP framework by considering time-dependent quantum states, time-dependent outcome sets, or multiple and entangled underlying states. Another possible future direction is to extend our proposed framework by assuming partial state observability in EA-MDP or using non-stationary rewards.

### Broader Impact Statement

This paper presents work whose goal is to advance the field of Machine Learning. There are many potential societal consequences of our work, none of which we feel must be specifically highlighted here.

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

# Supplementary Materials

## A  Proofs

This section presents the detailed proofs of the theorems in the paper.

### A.1  Proof of Theorem 1

Based on the definition of $V_\pi^{\tilde{\boldsymbol{s}}}(s)$, we can write

$$
\begin{aligned}
V_\pi^{\tilde{\boldsymbol{s}}}(s) &\doteq \mathbb{E}_\pi\Big[G_t \mid \tilde{S}_t = \tilde{\boldsymbol{s}}(s)\Big] \\
&= \mathbb{E}_\pi\Big[\tilde{r}_{t+1} + \gamma G_{t+1} \mid \tilde{S}_t = \tilde{\boldsymbol{s}}(s)\Big] \\
&= \sum_{\tilde{a}\in\tilde{\mathcal{A}}} \pi\big(\tilde{a}|\tilde{\boldsymbol{s}}(s)\big) \sum_{s'\in\mathcal{S}} \sum_r p\big(\tilde{\boldsymbol{s}}(s'),r|\tilde{\boldsymbol{s}}(s),\tilde{a}\big)\Big[r + \gamma\mathbb{E}_\pi\big[G_{t+1}|\tilde{S}_{t+1} = \tilde{\boldsymbol{s}}(s')\big]\Big].
\end{aligned}
\tag{26}
$$

The environment provides the reward $r = \tilde{r}(\tilde{\omega})$ to a possible outcome $\tilde{\omega}$ in each measurement on the state $|\tilde{s}'\rangle$. By substituting Equation (12) into the above equation, we get

$$
V_\pi^{\tilde{\boldsymbol{s}}}(s) = \sum_{\tilde{a}\in\tilde{\mathcal{A}}} \pi\big(\tilde{a}|\tilde{\boldsymbol{s}}(s)\big) \sum_{s'\in\mathcal{S}} \sum_{\tilde{\omega}\in\tilde{\Omega}} p\big(\tilde{\boldsymbol{s}}(s')|\tilde{\boldsymbol{s}}(s),\tilde{a}\big) P_{\tilde{\omega}}\big(\tilde{\boldsymbol{s}}(s')\big)\Big[\tilde{r}(\tilde{\omega}) + \gamma\mathbb{E}_\pi\big[G_{t+1}|\tilde{S}_{t+1} = \tilde{\boldsymbol{s}}(s')\big]\Big].
\tag{27}
$$

One measurement is not enough because it only gives one possible outcome from the quantum state's probability distribution. In this way, just one measurement does not provide any information about the quantum state of the environment. That is why it is essential to perform numerous measurements on the quantum state $|\tilde{s}'\rangle$ and calculate the expectation value of the reward operator. This expectation value is the reward function of the agent in EA-MDP. By using Equation (8) and sum over $\tilde{\omega}$ we have

$$
V_\pi^{\tilde{\boldsymbol{s}}}(s) = \sum_{\tilde{a}\in\tilde{\mathcal{A}}} \pi\big(\tilde{a}|\tilde{\boldsymbol{s}}(s)\big) \sum_{s'\in\mathcal{S}} p\big(\tilde{\boldsymbol{s}}(s')|\tilde{\boldsymbol{s}}(s),\tilde{a}\big)\Big[r\big(\tilde{\boldsymbol{s}}(s')\big) + \gamma V_\pi^{\tilde{\boldsymbol{s}}}(s')\Big].
\tag{28}
$$

The recursion for $Q_\pi^{\tilde{\boldsymbol{s}}}(s,\tilde{a})$ can be generated in a similar manner. Furthermore, it is important to mention that the relation between the value function $V_\pi^{\tilde{\boldsymbol{s}}}(s)$ and the action-value function $Q_\pi^{\tilde{\boldsymbol{s}}}(s,\tilde{a})$ is given by

$$
V_\pi^{\tilde{\boldsymbol{s}}}(s) = \sum_{\tilde{a}\in\tilde{\mathcal{A}}} \pi\big(\tilde{a}|\tilde{\boldsymbol{s}}(s)\big) Q_\pi^{\tilde{\boldsymbol{s}}}(s,\tilde{a}).
\tag{29}
$$

### A.2  Proof of Theorem 2

We need to show that the Bellman operator $T$ is a contraction with respect to the supremum norm $\|.\|_\infty$. Let us consider $V_1^{\tilde{\boldsymbol{s}}}$ and $V_2^{\tilde{\boldsymbol{s}}}$ as any value functions. The difference between these value functions after applying the Bellman operator $T$ to them is given by

$$
\big\|TV_1^{\tilde{\boldsymbol{s}}} - TV_2^{\tilde{\boldsymbol{s}}}\big\|_\infty = \max_{s\in\mathcal{S}} \big|\big(TV_1^{\tilde{\boldsymbol{s}}}\big)(s) - \big(TV_2^{\tilde{\boldsymbol{s}}}\big)(s)\big|.
\tag{30}
$$

Let us consider the Bellman operator for a value function as

$$
\begin{aligned}
\big(TV_1^{\tilde{\boldsymbol{s}}}\big)(s) &= \max_{\tilde{a}\in\tilde{\mathcal{A}}} \Bigg[\sum_{s'\in\mathcal{S}} p\big(\tilde{\boldsymbol{s}}(s')|\tilde{\boldsymbol{s}}(s),\tilde{a}\big)\Big(r\big(\tilde{\boldsymbol{s}}(s')\big) + \gamma V_1^{\tilde{\boldsymbol{s}}}(s')\Big)\Bigg], \\
\big(TV_2^{\tilde{\boldsymbol{s}}}\big)(s) &= \max_{\tilde{a}\in\tilde{\mathcal{A}}} \Bigg[\sum_{s'\in\mathcal{S}} p\big(\tilde{\boldsymbol{s}}(s')|\tilde{\boldsymbol{s}}(s),\tilde{a}\big)\Big(r\big(\tilde{\boldsymbol{s}}(s')\big) + \gamma V_2^{\tilde{\boldsymbol{s}}}(s')\Big)\Bigg].
\end{aligned}
\tag{31}
$$

The absolute difference is given by

$$\left| \left(TV_1^{\tilde{\mathbf{s}}}\right)(s) - \left(TV_2^{\tilde{\mathbf{s}}}\right)(s) \right| \leq \gamma \max_{\tilde{a} \in \tilde{A}} \sum_{s' \in \mathcal{S}} p\left(\tilde{\mathbf{s}}(s')|\tilde{\mathbf{s}}(s), \tilde{a}\right) \left| V_1^{\tilde{\mathbf{s}}}(s') - V_2^{\tilde{\mathbf{s}}}(s') \right|. \tag{32}$$

Since $\sum_{s' \in \mathcal{S}} p\left(\tilde{\mathbf{s}}(s')|\tilde{\mathbf{s}}(s), \tilde{a}\right) = 1$ and $0 \leq \gamma < 1$, we have

$$\left\| TV_1^{\tilde{\mathbf{s}}} - TV_2^{\tilde{\mathbf{s}}} \right\|_\infty \leq \gamma \max_{s' \in S} \left| V_1^{\tilde{\mathbf{s}}}(s') - V_2^{\tilde{\mathbf{s}}}(s') \right|. \tag{33}$$

Taking the supremum over all states $s$ leads to

$$\left\| TV_1^{\tilde{\mathbf{s}}} - TV_2^{\tilde{\mathbf{s}}} \right\|_\infty \leq \gamma \left\| V_1^{\tilde{\mathbf{s}}} - V_2^{\tilde{\mathbf{s}}} \right\|_\infty. \tag{34}$$

Since $0 \leq \gamma < 1$, this inequality shows that $T$ is a contraction mapping.

### A.3  Proof of Theorem 3

Given that $T$ is a contraction mapping, the Banach fixed-point theorem ensures that $T$ has a fixed point $V_{\pi^*}^{\tilde{\mathbf{s}}}$, which satisfies the following condition

$$TV_{\pi^*}^{\tilde{s}} = V_{\pi^*}^{\tilde{s}}. \tag{35}$$

The fixed point $V_{\pi^*}^{\tilde{s}}$ of the Bellman operator $T$ is the optimal value function that satisfies the Bellman optimal equation. $V_{\pi^*}^{\tilde{s}}$ is the optimal value function. The existence of the fixed point $V_{\pi^*}^{\tilde{s}}$ means that there exists an optimal policy $\pi^*$ that achieves this value function. Therefore, the EA-MDP has at least one optimal policy. Given the optimal value function $V_{\pi^*}^{\tilde{s}}$, we define the corresponding optimal policy $\pi^*$ as

$$\pi^*\left(\tilde{\mathbf{s}}(s)\right) = \arg\max_{\tilde{a} \in \tilde{A}} \left[ \sum_{s' \in \mathcal{S}} p\left(\tilde{\mathbf{s}}(s')|\tilde{\mathbf{s}}(s), \tilde{a}\right) \left( r\left(\tilde{\mathbf{s}}(s')\right) + \gamma V^{\tilde{\mathbf{s}}}(s') \right) \right]. \tag{36}$$

### A.4  Proof of Theorem 4

Let $\hat{V}^{\tilde{\mathbf{s}}}$ denote the value function corresponding to the empirical estimate of the reward $\hat{r}$. To prove the theorem, we proceed in three steps. In the first step, we derive a bound between $\hat{r}$ and $r$. In the second step, we establish a bound between $\hat{V}^{\tilde{\mathbf{s}}}$ and $V^{\tilde{\mathbf{s}}}$ under the same policy. In the final step, we combine these results to determine the  bound.
**Step 1**: Let $X$ be the reward obtained from a single measurement of $\tilde{s}(s')$. The empirical estimate of the expected reward from $n$ independent measurements is $\hat{r}_n(\tilde{s}(s')) = \frac{1}{n} \sum_{i=1}^n X_i$. To estimate the deviation between the empirical mean $\hat{r}(\tilde{s}(s'))$ and the true expectation value of reward $r = \mathbb{E}[X] = \langle \psi | \mathbf{R} | \psi \rangle$, we apply the Hoeffding inequality (Hoeffding (1963); Takagi et al. (2022)) for bounded random variables. For any $\varepsilon \geq 0$, we have

$$\Pr(|\hat{r}(\tilde{s}(s')) - r(\tilde{s}(s'))| \geq \varepsilon) \leq 2 \exp\left( -\frac{2n\varepsilon^2}{d^2(\tilde{s}(s'))} \right), \tag{37}$$

where $d(\tilde{s}(s'))$ denotes the range of possible measurement outcomes for the state $\tilde{s}(s')$. By setting $\varepsilon$ as the absolute error tolerance for reward estimation and letting $\delta$ be the failure probability (so that the bound holds with probability at least $1 - \delta$), we obtain

$$|\hat{r}(\tilde{s}(s')) - r(\tilde{s}(s'))| \leq d(\tilde{s}(s')) \sqrt{\frac{\log(2/\delta)}{2n}}. \tag{38}$$

Taking the supremum over all states yields

$$\|\hat{r} - r\|_\infty \leq \Delta \sqrt{\frac{\log(2/\delta)}{2n}}, \tag{39}$$

where $\Delta$ is the maximum possible value of $d$ and is the difference between the largest and smallest eigenvalues of $\mathbf{R}$. We used the fact that the expectation value of $\mathbf{R}$ is always bounded between its largest and smallest eigenvalues.

**Step 2.** For a fixed policy $\pi$ and state $\tilde{\boldsymbol{s}}(s)$, using Equation (13a), we have

$$V_\pi^{\tilde{\boldsymbol{s}}}(s) - \hat{V}_\pi^{\tilde{\boldsymbol{s}}}(s) = \mathbb{E}_\pi \left[ \sum_{k=0}^\infty \gamma^k \left( r_{t+k+1} - \hat{r}_{t+k+1} \right) \,\middle|\, \tilde{S}_t = \tilde{\boldsymbol{s}}(s) \right]. \tag{40}$$

By linearity of expectation, taking the supremum over all states and applying Equation (39), we obtain

$$\left\| V_\pi^{\tilde{\boldsymbol{s}}} - \hat{V}_\pi^{\tilde{\boldsymbol{s}}} \right\|_\infty \leq \frac{\Delta}{1-\gamma} \sqrt{\frac{\log(2/\delta)}{2n}} \tag{41}$$

**Step 3.** For the difference between the true optimal value function and the optimal value function computed from finitely many measurements, we have

$$V_{\pi^*}^{\tilde{\boldsymbol{s}}}(s) - V_{\hat{\pi}^*}^{\tilde{\boldsymbol{s}}}(s) = (V_{\pi^*}^{\tilde{\boldsymbol{s}}}(s) - \hat{V}_{\pi^*}^{\tilde{\boldsymbol{s}}}(s)) + (\hat{V}_{\pi^*}^{\tilde{\boldsymbol{s}}}(s) - \hat{V}_{\hat{\pi}^*}^{\tilde{\boldsymbol{s}}}(s)) + (\hat{V}_{\hat{\pi}^*}^{\tilde{\boldsymbol{s}}}(s) - V_{\hat{\pi}^*}^{\tilde{\boldsymbol{s}}}(s))$$
$$\leq (V_{\pi^*}^{\tilde{\boldsymbol{s}}}(s) - \hat{V}_{\pi^*}^{\tilde{\boldsymbol{s}}}(s)) + (\hat{V}_{\hat{\pi}^*}^{\tilde{\boldsymbol{s}}}(s) - V_{\hat{\pi}^*}^{\tilde{\boldsymbol{s}}}(s)), \tag{42}$$

where we used the fact that $\hat{\pi}^*$ is optimal with respect to $\hat{r}$, and therefore $(\hat{V}_{\pi^*}^{\tilde{\boldsymbol{s}}} - \hat{V}_{\hat{\pi}^*}^{\tilde{\boldsymbol{s}}}) \leq 0$. Taking absolute values, applying the triangle inequality, and using the supremum norm together with Equation (41), we obtain the result in Equation (17).

## B  EA-QL Algorithm

In this section, we suggest Algorithm 1 to address the epsilon-greedy Q-learning algorithm by including aspects of epistemic ambivalence. We propose an EA-epsilon-greedy strategy that includes a measure of EA. By considering the EA linked with each action's Q-value, the agent can make more informed decisions in a given state.

---

**Algorithm 1** EA-Epsilon-greedy Q-Learning algorithm

---

**Require:** $\alpha$: learning rate, $\gamma$: discount factor, $\varepsilon$: a small number
  Initialize $Q^{\tilde{\boldsymbol{s}}}(s, \tilde{a})$ arbitrarily
  **for** each episode **do**
    Initialize underlying state $s$ and quantum state $\tilde{\boldsymbol{s}}(s)$
    **repeat** (each step of the episode)
      $\tilde{a} \leftarrow$ choose-action $(Q, \tilde{\boldsymbol{s}}(s), \varepsilon)$
      take action $\tilde{a}$ and evolve $s$ to $s'$ and $\tilde{\boldsymbol{s}}(s)$ to $\tilde{\boldsymbol{s}}(s')$
      ▷ *Calculate the reward using the quantum state $\tilde{\boldsymbol{s}}(s')$* ◁
      calculate $r(\tilde{\boldsymbol{s}}(s'))$ from quantum state $\tilde{\boldsymbol{s}}(s')$
      $Q^{\tilde{\boldsymbol{s}}}(s, \tilde{a}) \leftarrow Q^{\tilde{\boldsymbol{s}}}(s, \tilde{a}) + \alpha[r(\tilde{\boldsymbol{s}}(s')) + \gamma \max_{\tilde{a}'} Q^{\tilde{\boldsymbol{s}}}(s', \tilde{a}') - Q^{\tilde{\boldsymbol{s}}}(s, \tilde{a})]$
      $s \leftarrow s'$ and $\tilde{\boldsymbol{s}}(s) \leftarrow \tilde{\boldsymbol{s}}(s')$
    **until** $s$ is terminal
  **return** The $Q$-table that contains $Q^{\tilde{\boldsymbol{s}}}(s, \tilde{a})$ to determine the optimal policy $\pi^*$

---

## C  Reward Operator

**Theorem 6.** *Given an environment that provides an outcome reward function $\tilde{r} : \tilde{\Omega} \to \mathbb{R}$ for each projective measurement outcome $|\tilde{\omega}\rangle \in \tilde{\Omega}$, the reward operator $\mathbf{R}$ can be expressed as*

$$\boldsymbol{R} = \sum_{\tilde{\omega} \in \tilde{\Omega}} \tilde{r}(\tilde{\omega}) |\tilde{\omega}\rangle \langle\tilde{\omega}| . \tag{43}$$

**Proof:** In a projective measurement, the operator $\boldsymbol{P}_{\tilde{\omega}}$ is a Hermitian projection operator that can be expressed as $\boldsymbol{P}_{\tilde{\omega}} = |\tilde{\omega}\rangle \langle\tilde{\omega}|$. The probability of measuring the outcome $\tilde{\omega}$ when the system is in the quantum state $\tilde{s}(s')$ can be calculated using Equation (6) as follows

$$P_{\tilde{\omega}}\big(\tilde{\boldsymbol{s}}'(s')\big) = \langle \tilde{s}' \,|\, \boldsymbol{P}_{\tilde{\omega}} \,|\, \tilde{s}'\rangle = \langle \tilde{s}' \,|\, \tilde{\omega}\rangle \langle \tilde{\omega} \,|\, \tilde{s}'\rangle = |\langle \tilde{\omega} \,|\, \tilde{s}'\rangle|^2. \tag{44}$$

The expectation value of the reward can be calculated using Equation (8), which gives

$$\begin{aligned}
r\big(\tilde{\boldsymbol{s}}(s')\big) &= \sum_{\tilde{\omega}} P_{\tilde{\omega}}\big(\tilde{\boldsymbol{s}}(s')\big)\tilde{r}(\tilde{\omega}) \\
&= \sum_{\tilde{\omega}} \tilde{r}(\tilde{\omega}) \,|\langle \tilde{\omega} \,|\, \tilde{s}'\rangle|^2 \\
&= \sum_{\tilde{\omega}} \tilde{r}(\tilde{\omega}) \,\langle \tilde{s}' \,|\, \tilde{\omega}\rangle \langle \tilde{\omega} \,|\, \tilde{s}'\rangle \\
&= \langle \tilde{s}'| \left( \sum_{\tilde{\omega}} \tilde{r}(\tilde{\omega}) \,|\tilde{\omega}\rangle \langle \tilde{\omega}| \right) |\tilde{s}'\rangle \\
&= \langle \tilde{s}' \,|\, \boldsymbol{R} \,|\, \tilde{s}'\rangle \\
&= \langle \boldsymbol{R} \rangle_{\tilde{s}'},
\end{aligned} \tag{45}$$

where $\boldsymbol{R}$ represents the reward operator, which is expressed as

$$\boldsymbol{R} = \sum_{\tilde{\omega}} \tilde{r}(\tilde{\omega}) \,|\tilde{\omega}\rangle \langle \tilde{\omega}|. \tag{46}$$

The reward operator in Theorem 6 is an observable quantity used to calculate the expectation value of reward within the quantum mechanical framework. It makes a connection between each outcome $|\tilde{\omega}\rangle$ and its corresponding reward $\tilde{r}(\tilde{\omega})$ by projecting the quantum state onto the outcome $|\tilde{\omega}\rangle$.

## D   Separated Outcomes in EA-MDP

In Section 4, we demonstrated the EA-MDP theorems for general outcome sets $|\tilde{\omega}\rangle$, which may include multiple underlying states simultaneously. While formulating the quantum state $\tilde{\mathbf{s}}(\mathbf{s})$, we considered quantum states that include only a single underlying state $s$, as shown in Equation (3). Therefore, it is preferable to consider only one underlying state in the measurement. In this case, the outcome $|\tilde{\omega}\rangle$ is separable into two independent components. The first component is used to measure the underlying state $s''$, while the other component is used to measure the EA quantum state $|\omega^{(\text{EA})}\rangle \in \mathcal{H}_{\text{EA}}$. In other words, $|\tilde{\omega}\rangle = |s''\rangle \otimes |\omega^{(\text{EA})}\rangle$. The finite set of measurement outcomes in EA component is $\Omega^{(\text{EA})} = \{|\omega^{(\text{EA})}\rangle\}$. We use $\tilde{r}(\tilde{\omega}) = \tilde{r}(s'', \omega^{(\text{EA})})$ to illustrate the separation between the underlying state and the EA outcome in the outcome reward function. We reformulate the reward operator given in Equation (43) using this separation as follows

$$\boldsymbol{R}^{(\text{sp})} = \sum_{s'', \omega^{(\text{EA})}} \tilde{r}(s'', \omega^{(\text{EA})})\big( |s''\rangle \langle s''| \big) \otimes \big( |\omega^{(\text{EA})}\rangle \langle \omega^{(\text{EA})}| \big), \tag{47}$$

where $\boldsymbol{R}^{(\text{sp})}$ is the reward operator with separated outcome set. The completeness relationship requires that

$$\sum_{s'' \in \mathcal{S}} |s''\rangle \langle s''| = \boldsymbol{I}_n, \qquad \sum_{\omega^{(\text{EA})} \in \Omega^{(\text{EA})}} |\omega^{(\text{EA})}\rangle \langle \omega^{(\text{EA})}| = \boldsymbol{I}_m, \tag{48}$$

where $\boldsymbol{I}_n$ is the identity operator with dimension $n$. In Equation (47), the outcome reward function $\tilde{r}(s', \omega^{(\text{EA})})$ depends on both $s'$ and $\omega^{(\text{EA})}$. The dependency of $s'$ could be transferred to one of the EA components of the next quantum states by introducing an additional dimension to the state.

**Assumption 3.** *We assume that the environment provides the reward to the EA component of the outcome in separated outcomes, $|\omega^{(EA)}\rangle$.*

This goal can be achieved simply by setting

$$\tilde{r}(s', \omega^{(\mathrm{EA})}) \equiv \tilde{r}(\omega^{(\mathrm{EA})}). \tag{49}$$

The expectation value of the reward, given the next quantum state $\tilde{\boldsymbol{s}}(s')$ and separated outcomes, is calculated as follows

$$r\big(\tilde{\boldsymbol{s}}(s')\big) = \Big\langle \boldsymbol{R}^{(\mathrm{sp})} \Big\rangle_{\tilde{s}'} = \Big\langle \tilde{s}' \,\Big|\, \boldsymbol{R}^{(\mathrm{sp})} \,\Big|\, \tilde{s}' \Big\rangle = \sum_{\omega^{(\mathrm{EA})}} \tilde{r}(\omega^{(\mathrm{EA})}) \Big| \Big\langle \omega^{(\mathrm{EA})} \,\Big|\, \psi_{s'} \Big\rangle \Big|^2. \tag{50}$$

This relation shows how the expectation value of the rewards depends on $\tilde{\boldsymbol{s}}(s')$ and the corresponding EA quantum state.

Under this assumption, the reward operator in Equation (47) for the separated outcomes is reduced to

$$\boldsymbol{R}^{(\mathrm{sp})} = \Big( \sum_{s''} |s''\rangle \langle s''| \Big) \otimes \Big( \sum_{\omega^{(\mathrm{EA})}} \tilde{r}(\omega^{(\mathrm{EA})}) |\omega^{(\mathrm{EA})}\rangle \langle \omega^{(\mathrm{EA})}| \Big) = \boldsymbol{I}_n \otimes \boldsymbol{R}^{(\mathrm{EA})}, \tag{51}$$

where $\boldsymbol{R}^{(\mathrm{EA})}$ is the EA reward operator, represented as

$$\boldsymbol{R}^{(\mathrm{EA})} = \sum_{\omega^{(\mathrm{EA})}} \tilde{r}(\omega^{(\mathrm{EA})}) |\omega^{(\mathrm{EA})}\rangle \langle \omega^{(\mathrm{EA})}|. \tag{52}$$

The EA reward operator $\boldsymbol{R}^{(\mathrm{EA})}$ allows for the measurement of the reward associated with the EA quantum states.

In order to calculate the overlap between each $|\omega^{(\mathrm{EA})}\rangle$ and the EA component of $\tilde{\boldsymbol{s}}(s')$, we expand each $|\omega^{(\mathrm{EA})}\rangle$ in terms of the EA basis states $|j\rangle$ using a linear map. Afterwards, by using the inner product of quantum states, we compute the overlap. Each basis $|j\rangle$ can be considered a piece of evidence, and when each outcome involves multiple bases, it indicates that rewards are given based on the presence of these pieces of evidence. When computing the quantum probability of an outcome, we are essentially determining how closely this outcome aligns with the quantum state. The interference effect can then either amplify or diminish this probability.

In a special case, the linear map is bijective, meaning that every member in the set $\Omega^{(\mathrm{EA})}$ has a one-to-one correspondence with a distinct element in the EA basis, denoted as $|\omega^{(\mathrm{EA})}\rangle = |j\rangle$. We have two more theorems for this bijective mapping.

**Theorem 7.** *A mapping $\boldsymbol{w} : \tilde{\mathcal{S}} \to [0,1]^m$ determines the model for each individual state. For a given quantum state $\tilde{\boldsymbol{s}} \in \tilde{\mathcal{S}}$, the vector $\boldsymbol{w}\big(\tilde{\boldsymbol{s}}(s)\big)$ specifies all ratios of the quantum state $|\tilde{s}\rangle$ being in all the EA quantum basis states $|j\rangle$ with underlying state $s$. In other words,*

$$\boldsymbol{w}\big(\tilde{\boldsymbol{s}}(s)\big)[j] = c_{s,j}^2 \qquad for \quad j = 0, 1, \ldots, m-1. \tag{53}$$

**Proof:** To determine the coefficient of the individual EA basis state, we calculate the overlap between states. $|s\rangle \otimes |j\rangle$ and the corresponding quantum state $\tilde{\boldsymbol{s}}(s)$. Therefore, the overlap, as introduced in Sec. 3, can be written as

$$
\begin{aligned}
\boldsymbol{w}\big(\tilde{\boldsymbol{s}}(s)\big)[j] &\doteq \mathbb{P}\Big[ |s\rangle \otimes |j\rangle \in \tilde{\boldsymbol{s}}(s) \Big], \quad for \quad j = 0, 1, \ldots, m-1 \\
&= \Big| \Big( \langle s| \otimes \langle j| \Big) \Big( |\tilde{\boldsymbol{s}}\rangle \Big) \Big|^2 \\
&= \Big| \Big( \langle s| \otimes \langle j| \Big) \Big( |s\rangle \otimes |\psi_s\rangle \Big) \Big|^2 \\
&= |\langle s \mid s \rangle \langle j \mid \psi_s \rangle|^2,
\end{aligned} \tag{54}
$$

where we used the mixed product property of the Kronecker product,

$$(\boldsymbol{A} \otimes \boldsymbol{B})(\boldsymbol{C} \otimes \boldsymbol{D}) = \boldsymbol{AC} \otimes \boldsymbol{BD}. \tag{55}$$

The state $|\psi_s\rangle$ is a linear combination of the EA basis states. By using Equation (4), we can proceed as follows

$$
\boldsymbol{w}\big(\tilde{\boldsymbol{s}}(s)\big)[j] = \left| \sum_{j'=0}^{m-1} c_{s,j'} \langle j \mid j' \rangle \right|^2 = \left| \sum_{j'=0}^{m-1} c_{s,j'} \delta_{j,j'} \right|^2 = |c_{s,j}|^2 , \tag{56}
$$

where we used orthogonality features between two quantum basis states $\langle j \mid j' \rangle = \delta_{j,j'}$ and $\delta_{j,j'}$ is the Kronecker delta, defined as

$$
\delta_{j,j'} = \left\{ \begin{array}{ll} 1 & j = j' \\ 0 & j \neq j' \end{array} \right. \tag{57}
$$

$|c_{s,j}|^2$ demonstrates the ratio of presence at state $s$ with EA basis $j$.

In our problem, $\boldsymbol{w}\big(\tilde{\boldsymbol{s}}(s)\big)[j]$ represents the probability of the quantum state $\tilde{\boldsymbol{s}}(s)$ being in the EA basis state $|j\rangle$ with the underlying state $|s\rangle$.

**Theorem 8.** *Let the environment provide outcome rewards for the EA outcomes $|\omega^{(EA)}\rangle \in \Omega^{(EA)}$, with a bijective mapping between $|\omega^{(EA)}\rangle$ and $|j\rangle$, such that $|\omega^{(EA)}\rangle = |j\rangle$. The expectation value of the reward operator with respect to $\tilde{\boldsymbol{s}}(s')$ is given by*

$$
r\big(\tilde{\boldsymbol{s}}(s')\big) = \sum_{j'} \tilde{r}(j') \boldsymbol{w}(\tilde{\boldsymbol{s}}(s'))[j']. \tag{58}
$$

**Proof:** To calculate the expectation value of the reward operator with Equation (50), we use Equation (2) which is explained in Section 3. The expectation value of the reward operator with respect to the quantum state $\tilde{\mathbf{s}}(s')$ can be calculated as follows:

$$
\begin{aligned}
r\big(\tilde{\boldsymbol{s}}(s')\big) &= \langle \boldsymbol{R} \rangle_{\tilde{\mathbf{s}}(s')} \\
&= \langle \tilde{\mathbf{s}}(s') \,|\boldsymbol{R}|\, \tilde{\mathbf{s}}(s') \rangle \\
&= \Big( \langle s'| \otimes \langle \psi_{s'}| \Big) \boldsymbol{R} \Big( |s'\rangle \otimes |\psi_{s'}\rangle \Big)
\end{aligned} \tag{59}
$$

By replacing the reward operator with the one defined in Equation (51), we obtain

$$
\begin{aligned}
r\big(\tilde{\boldsymbol{s}}(s')\big) &= \Big( \langle s'| \otimes \langle \psi_{s'}| \Big) \Big( \boldsymbol{I}_n \otimes \boldsymbol{R}^{(\mathrm{EA})} \Big) \Big( |s'\rangle \otimes |\psi_{s'}\rangle \Big) \\
&= \langle s' \,|\, \boldsymbol{I}_n \,|\, s' \rangle \Big\langle \psi_{s'} \,\Big|\, \boldsymbol{R}^{(\mathrm{EA})} \,\Big|\, \psi_{s'} \Big\rangle \\
&= \Big\langle \psi_{s'} \,\Big|\, \boldsymbol{R}^{(\mathrm{EA})} \,\Big|\, \psi_{s'} \Big\rangle ,
\end{aligned} \tag{60}
$$

where we used $\langle s' | \boldsymbol{I}_n | s' \rangle = \langle s' | s' \rangle = 1$. Assuming this theorem's proposition $|\omega^{(\mathrm{EA})}\rangle = |j\rangle$ and expanding $|\psi_{s'}\rangle$ in the basis $|j\rangle$, we obtain

$$
\begin{aligned}
r\left(\tilde{\boldsymbol{s}}(s')\right) &= \left( \sum_{j=0}^{m-1} c_{s',j}^* \langle j | \right) \left( \sum_{j'=0}^{m-1} \tilde{r}(j') |j'\rangle \langle j'| \right) \\
&\quad \times \left( \sum_{j''=0}^{m-1} c_{s',j''} |j''\rangle \right) \\
&= \sum_{j,j',j''} \tilde{r}(j') c_{s',j}^* c_{s',j''} \langle j | j' \rangle \langle j' | j'' \rangle \\
&= \sum_{j,j',j''} \tilde{r}(j') c_{s',j}^* c_{s',j''} \delta_{j,j'} \delta_{j',j''} \\
&= \sum_{j'} \tilde{r}(j') |c_{s',j'}|^2 \\
&= \sum_{j'} \tilde{r}(j') \boldsymbol{w}\left(\tilde{\boldsymbol{s}}(s')\right)[j'].
\end{aligned}
\tag{61}
$$

Each complex probability amplitude can be expressed as $c_{s,j} = r_{s,j} e^{i\theta_{s,j}}$, where $r_{s,j}$ and $\theta_{s,j}$ represent the magnitude and the phase, respectively. In a bijective mapping, the phase disappears, resulting in the expression $|c_{s,j}|^2 = r_{s,j}^2$. This implies that neither constructive nor destructive quantum interference is present. In this scenario, the reward is equivalent to the standard expected value of the reward in an MDP without any quantum mechanical features. In this case, we only calculate the probability of each piece of evidence.

# E  Compact representations of non-separable outcome sets

For fully non-separable outcome sets, a naive representation of $mn$ outcome quantum states in an $mn$-dimensional Hilbert space scales as $O((mn)^2)$ in memory and computation. In the Section D of the supplementary material, we detailed separable outcomes that reduce the complexity to $O(mn)$. Beyond the separable case of the outcome set, there are several structured parameterization strategies quantum mechanics and quantum information Anshu & Arunachalam (2024). They can keep computation practical even in non-separable setups. Below we briefly outline a few examples.

1. Sparse encodings Sherbert et al. (2022); Bellante & Zanero (2022): Each outcome $\omega_i \in \Omega$ has $k_i \ll mn$ nonzero entries. Storing $mn$ such outcomes then costs on the order of $O(\sum_{i=1}^{mn} k_i)$.

2. Low-dimensional parametric forms Lange et al. (2024): The components are generated via a parametric map $(\omega_i)_j = f(j; \boldsymbol{\theta}_i)$, where the dimension $p_i = \dim(\boldsymbol{\theta}_i)$ satisfies $p_i \ll mn$. In this case the memory requirement is reduced to $O(\sum_i p_i)$, or even $O(p)$ if many outcomes share a common parameter set.

3. Tensor factorizations Cirac et al. (2021): If the underlying space factors as a tensor product of $L = O(\log(mn))$ subsystems, each outcome can be represented as a tensor-network state with local dimension $d$ and bond dimension $\chi$. In this case, each outcome can be stored using $O(Ld\chi^2)$ parameters, which is (potentially) exponentially more efficient than $O(mn)$ in a fully dense representation of an outcome. In this case, the total storage of the outcome set scales as $O(mn(Ld\chi^2))$.

4. Operator-based descriptions Xu et al. (2023): Each outcome is represented as $|\omega_i\rangle = \mathbf{U}_i |\phi\rangle$, where $|\phi\rangle$ is a fixed reference state and $\mathbf{U}_i$ is a unitary operator with $L_i$ parameters. The storage cost then scales as $O(mn + \sum_i L_i)$, instead of storing all components explicitly.

5. Stabilizer-based encodings Sun et al. (2024): Each outcome is characterized as the common eigenvector of $O(L)$ commuting operators acting on $L = O(\log(mn))$ subsystems. This yields tableau-type descriptions scaling like $O(mnL^2)$ for $mn$ outcomes.

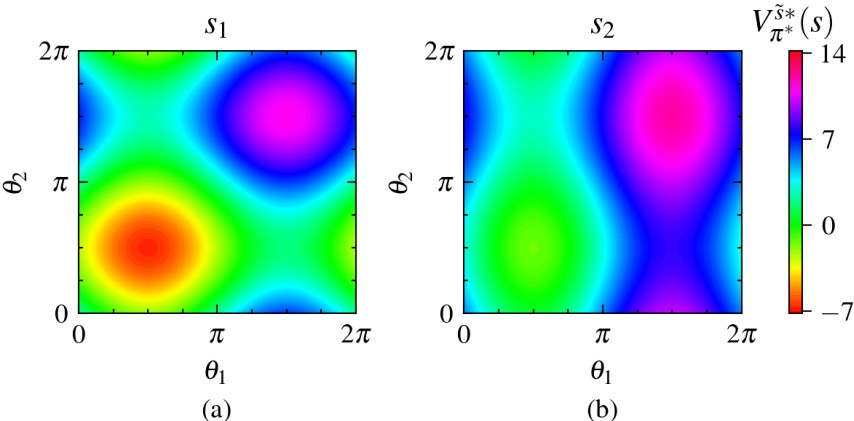

Figure 4: The optimal value function for the two-site example in the presence of EA is computed with parameters $\tilde{r} = (-1, 1, 2)$, $c_1 = (\frac{2}{3}, \frac{2}{3}e^{i\theta_1}, \frac{1}{3})$, and $c_2 = (\frac{2}{3}, \frac{1}{3}e^{i\theta_2}, \frac{2}{3})$. The set of outcomes is defined in Equation (21). (a) $V^{\tilde{s}*}(s_1)$, (b) $V^{\tilde{s}*}(s_2)$ for different values of $\theta_1$ and $\theta_2$.

In all of these methods, additional lossy compression is possible by truncating small coefficients, reducing ranks, or simplifying parameterizations, while keeping the approximation error under control. These modifications in the representation of the outcome set preserve the Bellman form, since they are only used to compute the reward via the reward operator.

# F  Additional Experimental Results

## F.1  Additional Experimental Results On Model 1: Two-site System

Complex probability amplitudes cause interference in quantum mechanics, leading to constructive or destructive interference patterns (increasing and decreasing the probability of the outcome). In Fig. 4, the impact of complex probability amplitude (using a factor $e^{i\theta}$, where $\theta$ is a phase factor) on the optimal value function is demonstrated. By employing probability amplitudes $c_1 = (\frac{2}{3}, \frac{2}{3}e^{i\theta_1}, \frac{1}{3})$ and $c_2 = (\frac{2}{3}, \frac{1}{3}e^{i\theta_2}, \frac{2}{3})$, we observe significant changes in the optimal value function as phase factors $\theta_1$ and $\theta_2$ are modified.

## F.2  Additional Experimental Results On Model 2: More Complex Example of Many-Site Systems

In Fig. 5, the optimal value function for different values of discount $\gamma$ and reward for $\phi_1 = \phi_2 = 0$ is shown. The contour plot shown in Fig. 6 displays the entire phase diagram of the optimal value function for parameters $\phi_1$ and $\phi_2$ within the range $[0, 2\pi]$.

To compare the performance of an EA-aware algorithm with a conventional RL-based algorithm, we run EA-QL and Soft Q-Learning algorithm (S-QL)(Eysenbach & Levine (2019); Cai et al. (2024); Haarnoja et al. (2017); Jeong & Lee (2024)) in our lattice environment. However, conventional RL-based agents are not aware of EA in the environment, and hence, are not capable of computing the expectation value of reward over outcomes with appropriate weights. To keep it fair, we assume that, similar to the EA-QL agent, the environment provides all possible outcome rewards, and the S-QL agent receives the average of the outcome rewards at the locations where EA exists. we consider outcome rewards to be $\tilde{r} = (-1, 1, -2, -4)$. The probability amplitudes and the set of outcomes are the same as Equation (24) and (25), with $\phi_1 = \phi_2 = 0$. The temperature parameter in softmax function is set to $\tau = 0.5$ and $\varepsilon = 0.1$. We run the algorithms for 7000 episodes and report the results in Fig. 7 for 50 independent runs. In Fig. 7-(a), we plot the average discounted cumulative rewards over episodes, which we refer to as *average gain* and define as

$$\mathcal{G}(s) = \frac{1}{n} \sum_{i=1}^{n} G_T^{(i)}(s), \tag{62}$$

where $G_T^{(i)}(s)$ is the discounted cumulative reward of episode $i$ starting from initial state $S$, $T$ is the final time-stamp of the episode, and $n$ is the total number of episodes.

In addition, we calculate the *conditional action entropy* for a trajectory, defined as

$$\mathcal{E}(\pi) = - \sum_{t=0}^{T} \pi(\tilde{a}_t | \tilde{s}(s_t)) \log \pi(\tilde{a}_t | \tilde{s}(s_t)). \tag{63}$$

Fig. 7-(b) depicts the conditional action entropy for each episode throughout our experiment. Smaller values of this entropy indicate a more deterministic decision-making during each episode. As we see, EA-QL algorithm achieves a lower entropy over time as the agent's policy converges. In an $\varepsilon$-greedy-based algorithm like ours, the exploration is controlled mainly by the parameter $\varepsilon$ and smaller values of $\varepsilon$ result in more deterministic policies. When $\varepsilon = 0$, and assuming that there are no ties, the policy becomes deterministic. However, at certain states, there may be ties, i.e., multiple actions with identical q-values, requiring the agent to select one arbitrarily, which in turn leads to a localized increase in entropy. On the other hand, the objective of S-QL agent is to maximize both the reward and the entropy of the trajectory simultaneously. As a result, it achieves a higher trajectory entropy compared to our algorithm as the episodes progress.

**Von Neumann entropy:** In addition to conditional action entropy, the Von Neumann entropy can also be calculated to measure the mixedness of a quantum state. In quantum mechanics, the Von Neumann entropy is defined as $S(\rho) = -\mathrm{Tr}(\rho \log \rho)$ ( Nielsen & Chuang (2010)), where $\rho$ is density matrix. Since in our proposed EA-MDP formulation we used a pure quantum state, the density matrix can be written as $\rho = |\tilde{s}\rangle \langle \tilde{s}|$. Due to the fact that pure states do not have any mixedness, the Von Neumann entropy is always zero in our system.

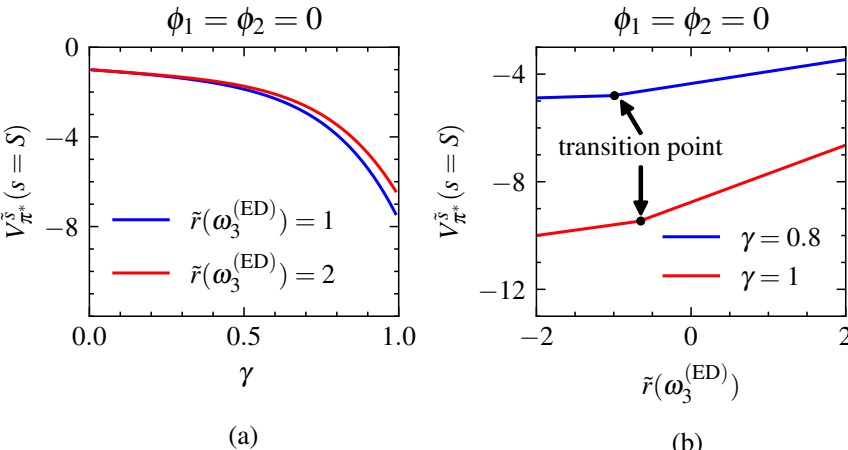

(a)

(b)

Figure 5: The optimal value function in the lattice with EA is computed using outcome rewards $\tilde{r} = (-1, -2, -3, \tilde{r}(\omega_3^{(EA)}))$ and probability amplitudes as presented in Equation (24). The set of outcomes is detailed in Equation (25), with $\phi_1 = \phi_2 = 0$. The optimal value function is shown in two cases: (a) for different values of $\gamma$ and (b) for different values of $\tilde{r}(\omega_3^{(EA)})$. At the transition point, the trajectory that maximizes rewards shifts to a new one.

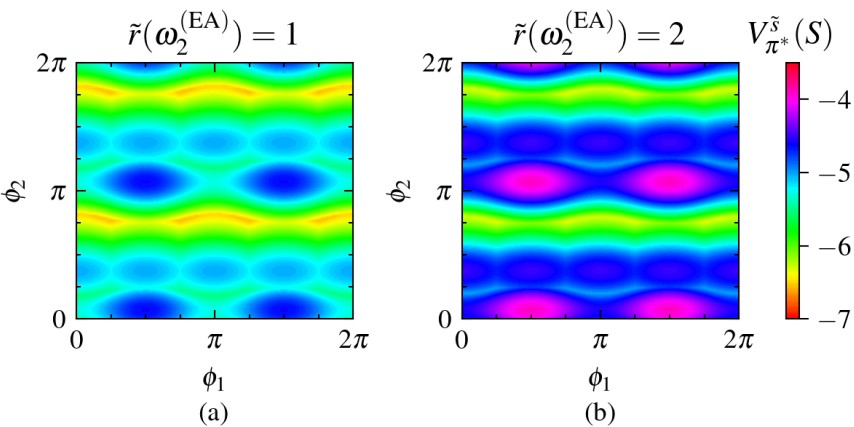

Figure 6: The optimal value function for the lattice example in the presence of EA, with outcome rewards $\boldsymbol{r} = (-1, -2, -3, \tilde{r}(\omega_2^{(EA)}))$, a discount factor $\gamma = 0.9$, probability amplitudes as presented in Equation (24), and the outcomes as listed in Equation (25). The effects are illustrated for (a) $\tilde{r}(\omega_2^{(EA)}) = 1$ and (b) $\tilde{r}(\omega_2^{(EA)}) = 2$, showing the oscillation of the optimal value function.

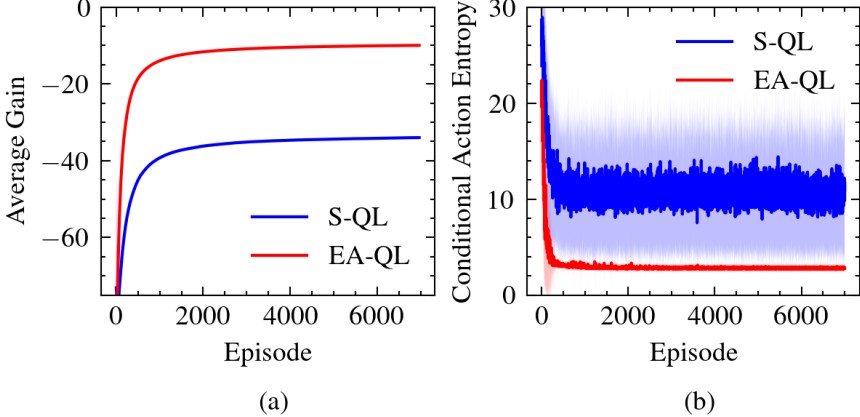

Figure 7: Comparison of performance of EA-QL and S-QL algorithms in the lattice environment. (a) Average discounted cumulative rewards over episodes. (b) Conditional action entropy over trajectories.

