# OpenReview forum: "Reinforcement Learning in the Presence of Epistemic Ambivalence"
_TMLR — Rejected by TMLR_

### Review · Reviewer_erHZ · 2025-10-24

**Summary Of Contributions:**

After carefully motivating the problem and situating it within the existing literature, the authors introduce the essential mathematical formalism derived from quantum mechanics. They then formally define the *Epistemically Ambivalent Markov Decision Process* (EA-MDP) framework by constructing its fundamental components: an augmented quantum state space that encodes epistemic ambivalence, a reward function expressed through quantum measurement operators, and a transition function describing the evolution of quantum-augmented states. The value function and action-value function are then formulated within this setting, leading to the derivation of the Bellman expectation equations. The theoretical section concludes with the definition of the Bellman optimality operator and a proof of its contraction property, establishing the existence of an optimal policy and value function in the EA-MDP framework.

Empirically, the authors validate the proposed model through two illustrative experiments: a two-site toy system and a multi-site lattice environment. These examples confirm the internal consistency of the theoretical derivations and demonstrate that the standard Q-learning algorithm can be extended to operate under epistemic ambivalence. Notably, the results show that epistemic ambivalence—encoded in the reward structure via complex probability amplitudes—can induce meaningful changes in the optimal policy, thereby supporting the central hypothesis that EA affects decision-making dynamics.

**Key strengths:**
- The paper introduces a novel and conceptually original framework bridging reinforcement learning and quantum mechanics.
- Modeling the reward function as an observable quantity, analogous to measurements in quantum systems, represents a particularly innovative and promising idea.
- The formalization is rigorous and mathematically consistent, with clear derivations of Bellman equations and convergence properties.
- The experiments, though simple, effectively illustrate the theoretical mechanisms and provide valuable intuition.

**Key weaknesses:**
- The framework requires converting classical data representations into quantum-like representations to effectively model the superposition principle, which may limit its immediate applicability in standard reinforcement learning settings.
- Constructing the set of measurement outcomes appears to be both challenging and crucial, as it directly determines how epistemic ambivalence is encoded into the model of decision-making problem.
- As the authors acknowledge, in the most general setting—particularly when measurement outcomes are non-separable—the proposed approach may not scale efficiently due to the size of the augmented state space, leading to computational and memory concerns.

**Audience:**

Yes

**Audience Explanation:**

I personally find that, despite the acknowledged challenges of modeling such behavior in real-world applications, the interpretation of the reward generation mechanism as analogous to the measurement process in quantum systems is particularly insightful. This perspective is both original and thought-provoking, and it deserves to be shared more widely to inspire further research in this direction and to promote the incorporation of this form of uncertainty into the broader Machine Learning—and especially the Reinforcement Learning—community.

**Broader Impact Concerns:**

Ethical concerns were sufficiently addressed in the papers's "Broader Impact Statement" section.

**Claims And Evidence:**

Yes

**Claims Explanation:**

The theoretical developments and experimental evaluation of the proposed framework, although conducted on toy examples, provide clear and convincing evidence that the approach is both sensible and theoretically well-grounded. The experiments demonstrate that the framework behaves as intended: it successfully encodes this specific notion of uncertainty, and the results clearly show how the agent’s decision-making process can be effectively influenced by the dependence of the reward signal on the underlying quantum state representation—and, consequently, by the interference effects emerging from the superposition of epistemic states. Finally, the authors are transparent in acknowledging the limitations of their approach, explicitly discussing its potential downsides and implementation challenges.

**Requested Changes:**

In my view, there appear to be a few minor typographical inconsistencies—for instance, in the presentation of the trajectory in the EA-MDP formulation, the subscripts seem to be incorrect. Similarly, in the Bellman equations, the multiplication symbol “$\times$” is used inconsistently across expressions. I also suspect a typo in the sentence: “EA occurs when multiple interpretations, explanations, or courses of action are possible, particularly in cases **where available when the data does** not clearly support one viewpoint over another.” Finally, in the experimental results section, I would recommend explicitly reporting the measurable set of outcomes (and their representations) to improve readability and facilitate comprehension of that part of the paper.

---

> ### Author Response · Authors · 2025-12-15
>
> We thank you for your helpful feedback. Below, we address your concerns and outline the changes we have made to the manuscript based on your comments. We hope that our responses and revisions meet your expectations.
>
> **Response to weaknesses:**
>
> - **(W1/W2)**
>
>     We agree with the reviewer that (i) converting classical information about epistemic ambivalence into a quantum-like superposition, and (ii) constructing an appropriate set of measurement outcomes, are both challenging tasks. Indeed, we explicitly acknowledged these limitations in the Challenges and Limitations section, where we highlighted the difficulties involved in mapping classical data to quantum states and in selecting a suitable outcome set to ensure meaningful and accurate reward computation.
>
>     We would also like to emphasize that one of the primary goals of our article is to introduce EA to the artificial intelligence community, particularly within reinforcement learning. We are therefore pleased that the reviewer found our work insightful and innovative. Our paper represents the first step in this direction, and to the best of our knowledge, no prior work has examined EA within machine learning. Hence, our contribution lays the groundwork for a new research direction in EA-MDP by demonstrating its potential for handling epistemic ambivalence. Consequently, there remains ample room for improvement and further development in this area, and we plan to address the challenges highlighted by the reviewer in our future work.
>
> - **(W3)**
>     In the most general case with non-separable outcome sets, a naive representation of the reward operator on a Hilbert space of dimension $mn$ requires $O((mn)^2)$ parameters and operations, reflecting the typical curse of dimensionality in quantum settings. Beyond the separable outcome case already discussed in the paper, several strategies can reduce this computational cost, including sparse encoding and variational approximations of the quantum state like using tensor factorizations. In the revised version, we will briefly mention these approaches in the supplementary material and provide appropriate references. We also include additional information on how the outcome set scales to larger non-separable outcome sets when such techniques are used. However, a systematic study of these methods is an interesting research direction in its own right and lies beyond the scope of the present paper.
>
> **Response to requested changes**
>
> - We corrected the trajectory subscripts and time indexing in the EA-MDP formulation and removed the unnecessary $\times$ notations from the Bellman equations. We fixed minor grammar issues, including the mentioned sentence. As requested, to improve readability, we moved the EA quantum states and measurable set of outcomes from the appendix into the main experiments section.

---

### Review · Reviewer_uNnH · 2025-11-03

**Summary Of Contributions:**

The paper introduces an epistemically ambivalent MDP (EA-MDP) that augments classical states with a quantum-inspired EA component to model persistent, conflicting evidence. Rewards are obtained via projective measurements and expectation over outcomes. The authors prove Bellman optimality, contraction, and existence of optimal policies, and present an EA-ɛ-greedy Q-learning algorithm validated on two toy domains to illustrate decision-making under ambivalence effects.

**Audience:**

Yes

**Audience Explanation:**

See **Strengths** above.

**Broader Impact Concerns:**

There is no impact concern in this paper.

**Claims And Evidence:**

Yes

**Claims Explanation:**

Strengths:

1. The EA-MDP construction is mathematically consistent: the state is lifted to a tensor-product Hilbert space, rewards are defined via a reward operator $R=\sum_\omega r(\omega)|\omega\rangle\langle\omega|$, and the expectation $\langle\tilde{s}| R|\tilde{s}\rangle$ plugs directly into a Bellman recursion, so the RL machinery still applies.

2. The paper proves that the EA Bellman operator
$(TV)(s)=\max_{\tilde{a}} \sum_{s^{\prime}} p\left(\tilde{s}^{\prime} \mid \tilde{s}, \tilde{a}\right)\left(r\left(\tilde{s}^{\prime}\right)+\gamma V\left(s^{\prime}\right)\right)$
is a $\gamma$-contraction, hence a fixed point and an optimal policy exist exactly as in classical discounted MDPs; this shows the quantum-inspired reward layer does not break standard dynamic programming theory.

3. The EA-ɛ-greedy Q-learning algorithm is a minimal modification of tabular Q-learning: it only changes how the immediate reward $r\left(\tilde{s}^{\prime}\right)$ is computed (via the outcome amplitudes), so convergence arguments can in principle reuse classical proofs under the stated assumptions.

**Requested Changes:**

Weaknesses:

1. The framework assumes time-independent EA states $|\psi_s\rangle$ for every classical $s$; this greatly simplifies the Bellman proofs but rules out the interesting case where ambivalence itself evolves and interferes with value propagation.

2. The reward-definition step relies on separable or at least well-structured outcome sets to keep the operator and its expectation tractable; for non-separable outcomes the paper notes the cost becomes $O\left((m n)^2\right)$, which weakens the claim of general applicability.

3. The proposed EA-ɛ-greedy algorithm does not exploit any specifically "quantum" property for exploration; it is essentially standard Q-learning with a more expensive reward oracle, so the algorithmic novelty is weaker than the modeling novelty.


Questions:

1. What changes in the contraction proof if the EA component $\left|\psi_s(t)\right\rangle$ is allowed to be time-varying, i.e. the transition also acts on the EA subspace?

2. Can the authors give an explicit bound that links approximation error in the reward operator $R$ (e.g. from finitely many measurements) to suboptimality of the learned value function?

3. For the non-separable outcome case, is there a sparse or low-rank parameterization of the outcome projectors that preserves the Bellman form while reducing the $O\left((m n)^2\right)$ storage/computation?

4. How would the EA-MDP definition interact with partial observability (POMDP-style belief states) does the EA layer sit on top of the belief, or does it replace it, and can the same existence/optimality proof be reused?

---

> ### Author Response · Authors · 2025-12-15
>
> We thank you for your insightful comments. We are glad that you found our theoretical framework coherent and our main theoretical claims convincing. Below, we address the identified weaknesses, respond to your questions, and outline the revisions made to the manuscript based on your comments.
>
> **Response to weaknesses and questions:**
>
> - **Response to (Q1/W1)**
>
> Allowing the EA quantum state to be time-dependent is certainly an interesting scenario. However, in our paper, we focus on keeping the EA-MDP formulation as simple as possible, as the current work aims to establish the foundation of EA in learning systems. Nevertheless, extending the current setting to a time-dependent quantum state is possible by introducing a unitary evolution operator such that $\ket{\psi_{s}(t_2)} = U(t_2, t_1)\ket{\psi_{s}(t_1)}$. We would also like to note that, in addition to quantum states, the current framework can be further extended to allow both the outcome reward function $\tilde{r}(\tilde \omega)$ and the measurement outcome $\ket{\tilde \omega}$ to be time-dependent. An important aspect of the quantum-inspired approach is that, even in the time-dependent setting, the expectation value of the reward remains bounded and smooth. Technically, the contraction and existence proofs do not fundamentally rely on time-independent EA quantum states. They only depend on the Markovian transition probabilities of the quantum states and the boundedness of the reward under a discount factor. As a result, the standard sup-norm contraction proof remains valid. In the revised manuscript, we will add a remark to clarify that Assumption 1 is being used as a modeling simplification, which is introduced to highlight the role of EA in the reward layer rather than a fundamental requirement for the Bellman contraction or optimality results in EA-MDP.
>
> - **Response to (Q2)**
>
> Let $\mathbf{R}$ denote the reward operator.  For any quantum state $\psi$, the expected reward $r(\psi) = \braket{\psi | \mathbf{R} | \psi}$ is bounded between the minimum and maximum eigenvalues of $\mathbf{R}$.  Let $X$ represent the outcome of a one-shot measurement, and let $\hat{r} = \tfrac{1}{n} \sum_{i=1}^{n} X_i$  denote the empirical estimate of the reward obtained from $n$ independent measurements.  Let $\hat{\pi}^\ast$ denote the optimal policy corresponding to this estimated reward. We can prove that with probability at least $1 - \delta$, the following bound holds
> $$\|V_{\hat\pi^\star}^{\tilde{\boldsymbol{s}}}-V_\star^{\tilde{\boldsymbol{s}}}\|_\infty
> \le \frac{2\Delta}{1 - \gamma}
> \sqrt{\frac{\log(2/\delta)}{2n}},
> $$
>
> where $\Delta$ denotes the difference between the maximum and minimum eigenvalues of the reward operator $\mathbf{R}$.  This bound quantifies the suboptimality of the learned policy $\hat{\pi}^*$ with respect to the true optimal policy.  The derivation of this bound relies on the Hoeffding inequality applied to bounded quantum measurement outcomes. We will include a theorem related to this bound in the revised version of the paper.

---

> > ### Author Response · Authors · 2025-12-15
> >
> > - **Response to (Q3/W2)**
> >
> >     For fully non-separable outcome sets, the naive representation of $mn$ outcome vectors in an $mn$-dimensional space scales as $O((mn)^2)$ in both memory and computation. Beyond the basic separable case of the outcome set that is already mentioned in the paper, there are several structured parameterization strategies in quantum mechanics and quantum information. They can keep the computation practical even in non-separable setups. Below we briefly outline a few examples.
> >
> >     - **Sparse encodings**:
> >     Each outcome $\omega_i \in \Omega$ has $k_i \ll mn$ nonzero entries. Storing $mn$ such outcomes then costs on the order of $O(\sum_{i=1}^{mn} k_i)$.
> >
> >     - **Low-dimensional parametric forms**:
> >     The components are generated via a parametric map $
> >     (\omega_i)_j = f(j; \boldsymbol{\theta}_i),
> >     $
> >     where the dimension $p_i = \dim(\boldsymbol{\theta}_i)$ satisfies $p_i \ll mn$.
> >     In this case the memory requirement is reduced to $O(\sum_i p_i)$, or even $O(p)$ if many outcomes share a common parameter set.
> >
> >     - **Tensor factorizations**:
> >     If the underlying space factors as a tensor product of $L = O(\log (mn))$ subsystems, each outcome  can be represented as a tensor-network state with local dimension $d$ and bond dimension $\chi$. In this case, each outcome can be stored using $O(L d \chi^2)$ parameters, which is (potentially) exponentially more efficient than $O(mn)$ in a fully dense representation of an outcome. In this case, the total storage of the outcome set scales as $O(mn (L d \chi^2))$.
> >
> >     - **Operator-based descriptions**:
> >     Each outcome is represented as
> >     $
> >     \ket{\omega_i} = \mathbf{U}_i \ket{\phi},
> >     $
> >     where $\ket{\phi}$ is a fixed reference state and $\mathbf{U}_i$ is an unitary operator with $L_i$ parameters. The storage cost then scales as $O(mn + \sum_i L_i)$, instead of storing all components explicitly.
> >
> >     - **stabilizer-based encodings**:
> >     Each outcome is characterized as the common eigenvector of $ O(L)$ commuting operators acting on $L = O(\log (mn))$ subsystems.  This yields tableau-type descriptions scaling like $O(mn L^2)$ for $mn$ outcomes.
> >
> >     In all of these methods, additional lossy compression is possible by truncating small coefficients, reducing ranks, or simplifying parameterizations, while keeping the approximation error under control. These modifications in the representation of the outcome set preserve the Bellman form, since they are only used to compute the reward via the reward operator.
> >     Since the general reader of the paper may not be familiar with these advanced methods, we will only mention them briefly in the  supplementary material and refer them in the discussion of the revised draft and provide appropriate references. In addition, a systematic exploration of these methods for large and non-separable outcome sets in EA-MDP is an interesting direction, but is beyond the scope of this paper.
> >
> > - **Response to  (W3)**
> >     We would like to emphasize that our paper represents the first step in introducing EA to machine learning problems, particularly within reinforcement learning. Accordingly, the main contribution of this work lies primarily in the modeling of the EA-MDP, rather than in algorithmic innovation. Specifically, our primary goal is to demonstrate that EA-MDPs can be integrated with existing RL methods and to empirically show how the EA reward layer alone can qualitatively influence learned policies. The EA–$\varepsilon$-greedy Q-learning algorithm is intentionally designed as a minimal adaptation of standard tabular Q-learning; the update rule and $\varepsilon$-greedy exploration remain classical, with the only modification being that the reward is computed via the EA reward operator $\mathbf{R}$, which is more expensive to evaluate, as correctly pointed out by the reviewer.
> >     We would also like to add that there is significant room for developing more EA-aware exploration. For example: (1) introducing exploration methods that actively probe directions in the EA state space and the outcomes to find the higher rewards. (2) using methods to gain more information of outcome sets and reward distribution encoded in $\mathbf{R}$ more easily. (3) finding approaches that exploit the geometry of the outcomes (e.g., amplitudes and phases) rather than relying  on the expectation value of the rewards.

---

> ### Author Response · Authors · 2025-12-15
>
> - **Response to  (Q4)**
>
> Our model is fundamentally different from PO-MDP, as in our model, the reward is generated based on some probability distribution over outcomes. First, while PO-MDP concentrates on situations with partial knowledge, EA-MDP employs quantum-inspired approaches for expressing and managing EA caused by contradicting evidence.
> Second, the final $\tilde{s}$ is the Kronecker product of $s$ and $\psi$, so we cannot really separate these two components in quantum state $\tilde{s}$ without any extra measurements.
> Third, in our approach, as shown in proof of Theorem 1, the environment needs to estimate the probability of outcomes and, consequently, estimate the expected reward (8), in order to compute the expectation value of the reward. An environment in a PO-MDP is not aware of such probabilities (is not aware of EA), hence PO-MDP methods fail in our setting.

---

### Review · Reviewer_R9tA · 2025-12-15

**Summary Of Contributions:**

The paper claims to "exploit the analogy between quantum mechanic states and EA to develop a theoretical framework for decision-making under EA and a policy".

**Audience:**

No

**Audience Explanation:**

I am afraid that I do not understand the purpose of the paper sufficiently to recommend acceptance.

There indeed seems to exist a literature on applying ideas of quantum mechanics to RL. But I do not really understand what concrete problem of "standard" RL they are trying to address, and the examples in Section 6 did not help me.

It is not very clear to me if quantum information is used here only as a mathematical formalism to analyze "normal" MDP with uncertainty on the parameters or on observations, or if the authors analyze MDPs over quantum systems.

-> I would ask a rewriting of the article that should make it convincing to a "classical" RL audience

**Broader Impact Concerns:**

No problem on that aspect

**Claims And Evidence:**

No

**Claims Explanation:**

The paper claims to "exploit the analogy between quantum mechanic states and EA to develop a theoretical framework for decision-making under EA and a policy"

**Requested Changes:**

At least, the authors should take a simple (classical) toy MDP, with an example of EA, and compare a classical treatment of the problem with their quantum solution.

Some minor comments:
p.2. l.3 "Ambivalence differs from uncertainty in that it persists even after the information becomes available" j

I do not understand how ambivalence differs from "aleatoric uncertainty" ; similarly, later I do not see the oppposition between aleatoric uncertainty and epistemic ambivalence

Somewhat below: I do not understand "particularly in cases where available when the data does not clearly support one viewpoint over another"

Section 2 "state of the art" : I do not see any connection with some references like "Zhang et al. (2020) studied the vulnerability of Deep Reinforcement Learning (DRL) agents to adversarial attacks on state observations"

p.4 strange to denote the module of a complex number c as \|c\| and not |c|

p.4 what does "The unitary operators O † O = OO † = I (where I is an identity operator), correspond to the evolution of the quantum states" mean?

Otherwise, the short recap on QIT is rather informative.

Section 4 p.4 "where H_S is the Hilbert space without any EA"  but what is H_S? not a standard notion in RL. The authors do not define a quantum MDP (even without EA).

p.5 "Note that, the quantum state" the ',' is to be removed

---

> ### Author Response · Authors · 2025-12-22
>
> We thank the reviewer for their valuable feedback. We would like to emphasize that our work introduces, for the first time, a reinforcement learning (RL) problem under epistemic ambivalence through the formulation of the EA-MDP. Hence, our paper is not addressing classic RL problems, but it solves the EA-MDP using the tools from classic RL and quantum mechanics. We emphasize that our paper uses quantum mechanics formalism as a tool to formulate and analyze a novel and extended MDP framework called EA-MDP. In addition, in page 3, we explained that
> "The algorithm proposed in this paper uses quantum-inspired principles within a fully classical framework to model EA and support decision-making. By simulating quantum-like behavior, it avoids quantum hardware constraints such as the no-cloning theorem, decoherence, and noise in the quantum computers."
>
> > At least, the authors should take a simple (classical) toy MDP, with an example of EA, and compare a classical treatment of the problem with their quantum solution.
>
> In Section Experimental Results and Analysis, we already consider a simple toy MDP with EA and run our EA–$\varepsilon$-greedy Q-learning algorithm on it to show that it converges to the optimal policy under EA, in line with our theoretical results. In response to your suggestion, we will additionally include a soft Q-learning baseline in the revision, applied to the same toy MDP, to further illustrate the difference between a classical treatment and the EA-MDP formulation.
>
>
> > Some minor comments: p.2. l.3 "Ambivalence differs from uncertainty in that it persists even after the information becomes available" j
>
> > I do not understand how ambivalence differs from "aleatoric uncertainty" ; similarly, later I do not see the oppposition between aleatoric uncertainty and epistemic ambivalence
>
> > Somewhat below: I do not understand "particularly in cases where available when the data does not clearly support one viewpoint over another"
>
> We never mentioned in the paper that there is an opposition between aleatoric uncertainty and epistemic ambivalence. However, epistemic ambivalence differs from the mentioned uncertainties. Aleatoric uncertainty corresponds to the randomness in random variables engaged in the experiment. Epistemic ambivalence corresponds to conflicting information received by the agent.
>
> In the revised manuscript, we edit the typo to "EA occurs when multiple interpretations, explanations, or courses of action are possible, particularly in cases where the available data does not clearly support one viewpoint over another."
>
>
> > Section 2 "state of the art" : I do not see any connection with some references like "Zhang et al. (2020) studied the vulnerability of Deep Reinforcement Learning (DRL) agents to adversarial attacks on state observations"
>
> Since no prior studies have considered this setting, i.e., epistemic ambivalence in reinforcement learning, we were unable to cite works that directly address our problem. Nevertheless, we discuss the closest related works, including the mentioned paper which have considered perturbation on state space.
>
> > p.4 strange to denote the module of a complex number c as |c| and not ||c||
>
> In the revised manuscript, we will standardize the notation for the modulus of a complex number to $|c|$.
>
> > p.4 what does "The unitary operators $O^\dagger O = OO^\dagger = I $ (where I is an identity operator), correspond to the evolution of the quantum states" mean?
> Otherwise, the short recap on QIT is rather informative.
>
> We aimed to emphasize that unitary operators represent norm-preserving transformations of quantum states, and are used to model the evolution of quantum states. We will rephrase this sentence accordingly in the revised version.
>
> > Section 4 p.4 "where $H_S$ is the Hilbert space without any EA" but what is $H_S$? not a standard notion in RL. The authors do not define a quantum MDP (even without EA).
>
> The $H_S$ is simply the Hilbert space spanned by the classical state set $\mathcal{S}$. Each classical state $s \in \mathcal{S}$ is embedded as a basis vector $\ket{s} \in  \mathcal{H}_{\mathcal{S}}$.  As mentioned in Eq. 3 and its following sentences, at each time step the agent receives
> $\ket{\tilde s}= \ket{s} \otimes \ket{\psi_s}$, where $\ket{s}$ is the underlying state.
> Generally speaking, the underlying state can serve as a classical description of certain aspects of the environment, such as restricting the agent to occupy only one location at each time step, which we considered in our experiments. This representation is crucial for integrating classical and quantum aspects within the framework of EA-MDPs.
>
> > p.5 "Note that, the quantum state" the ',' is to be removed
>
> Thank you for pointing this out. We have removed the comma in the revised version.

---

### Author Response · Authors · 2025-12-17
**Paper Revision Summary**

We thank the reviewers for their constructive and insightful feedback. We have revised the manuscript as advised. Below, we summarize the revisions made in the current version of the paper.

- We carefully proofread the manuscript to eliminate typographical errors and grammatical inconsistencies and to improve notational consistency.

- We added Theorem 4 that provides an explicit suboptimality bound for the learned policy under a finite number of measurements.

- We added a remark about how the framework can extend to
time-dependent EA quantum states.

- We moved the EA quantum states and the measurable outcome sets from the supplementary material into the main experimental section to improve readability.

- We added a new section in the supplementary material to discuss the compact representation of non-separable outcome sets that helps to reduce memory and computational complexity.

- We added additional experiments to compare the performance of our proposed algorithm with a soft Q-Learning algorithm in terms of average gain and conditional action entropy.

---

### Decision · Action_Editor_dUSp · 2026-03-03

**Recommendation:** Reject

**Audience:**

No

**Audience Explanation:**

Given the focus of TMLR on the computational and mathematical principles behind ML, there are probably some readers of TMLR that are interested by the derivation of Q-learning (or other RL-related notions) in quantum setups.

That said, as previously discussed, this is not the paper’s main focus, and the proposed approach for modeling uncertainty in reinforcement learning is unlikely to appeal to the typical TMLR readership.

**Claims And Evidence:**

No

**Claims Explanation:**

Quoting the authors, the main claim of the paper is that "[..]our work introduces, for the first time, a reinforcement learning (RL) problem under epistemic ambivalence through the formulation of the EA-MDP. Hence, our paper is not addressing classic RL problems, but it solves the EA-MDP using the tools from classic RL and quantum mechanics."

All reviewers agreed that the authors do describe a variant of Bellman recursions and of the Q-learning algorithm that rely on axioms of quantum mechanics. However, the majority of the reviewers considered that the claim that this quantum-inspired framework is appropriate to model epistemic ambivalence (EA) in RL (and possibly better than existing approaches) is not supported by evidence, neither of methodological or empirical nature.

For most reviewers, the authors’ characterization of epistemic ambivalence -along with their argument that traditional machine learning cannot address it, and their analogy between EA and quantum mechanical states- remained a postulate.